# MIRA: A Score for Conditional Distribution Accuracy and Model Comparison

Sammy Sharief [* 1 2 3 4]  Justine Zeghal [* 1 2 3]  Gabriel Missael Barco [1 2 3]  Pablo Lemos [5]  Yashar Hezaveh [1 2 3 6 7]
Laurence Perreault-Levasseur [1 2 3 6 7]

## Abstract

We introduce Mira, a sample-based score for assessing the accuracy of a candidate conditional distribution using only joint samples from the true data-generating process. Relying on the principle that distributions coincide if they assign equal probability mass to all regions, we derive an analytic expression for the Mira statistic, whose average defines the Mira score. This formulation further allows us to compute theoretical reference values and uncertainty estimates when the candidate distribution matches the true one. This framework enables model comparison by quantifying the alignment between the conditional distribution of a candidate model and the true data generating process. Consequently, Mira enables Bayesian model comparison through direct posterior validation, bypassing the challenging evidence computation. We demonstrate its effectiveness across several toy problems and Bayesian inference tasks.
Code: https://github.com/SammyS15/mira-score.

## 1. Introduction

Generative modeling, the task of learning an underlying distribution from a dataset of samples to generate new, synthetic samples, has made striking progress over recent years and has become a central tool in probabilistic modeling, especially in high dimensions. One particular class of such models has been of specific interest: conditional generative models, which allow learning conditional distribution $p(x \mid y)$ from joint data-label samples $(x, y) \sim p(x, y)$, with various applications ranging from computer vision (e.g. Gatys et al., 2016; Rombach et al., 2022; Voleti et al., 2022), medicine (e.g. Ying et al., 2019; Pan et al., 2025), drug discovery and protein synthesis (e.g. Jumper et al., 2021), for Bayesian inference in the context of physics (for instance in high-energy particle physics experiments, e.g., Bellagente et al., 2020; Diefenbacher et al., 2024; Pazos et al., 2025, or in astrophysics, e.g., Remy et al., 2023; Adam et al., 2022; Feng et al., 2023; Legin et al., 2024) or geosciences (e.g Andry et al., 2025; Savary et al., 2025; Rozet & Louppe, 2023), only to name a few. However, assessing the quality of samples produced by generative models, in particular conditional models, is key to guaranteeing their trustworthiness for application, especially in high-stakes applications such as AI safety (e.g. Somepalli et al., 2023; Hitaj et al., 2017) or healthcare (e.g. Pinaya et al., 2022; Fernandez et al., 2024; Tudosiu et al., 2024; Zhu et al., 2023b; Bluethgen et al., 2024), as well as other scientific applications where data analysis requires high accuracy in uncertainty quantification.

The question of how well these models approximate the true, underlying distribution has been the focus of much recent work. Proposed methods for marginal distribution typically fall into two classes: likelihood-based methods, which rely on evaluating the density of true data samples under the generative model (e.g. Theis et al., 2016; Nalisnick et al., 2019; Nowozin et al., 2016; Yazici et al., 2020; Le Lan & Dinh, 2021), and sample-based methods (e.g. Heusel et al., 2017; Salimans et al., 2016; Sajjadi et al., 2018; Stein et al., 2023; Alaa et al., 2022; Jiralerspong et al., 2023; Lemos et al., 2025), which rely on comparing true to generated samples. On the conditional distribution side, when only joint samples $(x, y) \sim p(x, y)$ are available, existing methods typically rely on coverage probability tests whose conclusions depend strongly on the choice of credible regions. Among these, TARP (e.g., Lemos et al., 2023) provides a powerful sufficient condition for accuracy. However, it does not readily support model ranking, as its coverage plots are harder to interpret quantitatively than Highest Posterior Density (HPD, Box & Tiao 1973; Hermans et al. 2022) expected coverage curves. Other approaches include probability integral transform methods such as Simulation-Based

---

[*]Equal contribution [1]Mila – Québec AI Institute, Montreal, Quebec, Canada [2]Ciela Institute, Montreal, Quebec, Canada [3]Université de Montréal, Montreal, Quebec, Canada [4]Université Paris-Saclay, Université Paris Cité, CEA, CNRS, AIM, 91191, Gif-sur-Yvette, France [5]SandboxAQ, Palo Alto, California, USA [6]CCA – Flatiron Institute, 162 5th Ave, New York, NY 10010, USA [7]Trottier Space Institute, Montreal, Quebec, Canada. Correspondence to: Sammy Sharief <sammy.sharief@universite-paris-saclay.fr>, Justine Zeghal <justine.zeghal@umontreal.ca>.

*Proceedings of the 43rd International Conference on Machine Learning*, Seoul, South Korea. PMLR 306, 2026. Copyright 2026 by the author(s).

Calibration (e.g., Gneiting et al., 2008; Talts et al., 2018), which do not easily scale to high dimensions, and classifier-based scores (e.g., Linhart et al., 2023), which require neural network training.

In this work, we introduce Mira (Mass In Random Areas), a joint-sample-based score for conditional distributions (validating the distributions for all conditioning variables). Building on the principle that two distributions are equal if they assign the same probability mass to all regions, we derive, under the null hypothesis, an analytic expression for the probability that a true sample falls within a region given the number of candidate samples in that region. Averaging this probability over many joint true samples $(x, y) \sim p(x, y)$ and regions yields the Mira score. This analytic formulation further enables us to derive key theoretical quantities, including the expected score and uncertainty under correctly specified distributions. This property makes Mira useful not only for quantifying conditional distribution accuracy, but also as a practical tool for Bayesian model comparison. By shifting the comparison from simulation space to parameter space, Mira provides a scalable alternative to evidence estimation, remaining effective in high-dimensional problems where classical approaches such as Bayes factors are often intractable (Kass & Raftery, 1995; Alsing et al., 2018; Spurio Mancini et al., 2023). Our contributions are:

1. We propose Mira, a framework for quantifying the accuracy of conditional distributions using only joint samples from the true model.

2. We establish theoretical quantities for the score and its uncertainty.

3. We demonstrate its effectiveness on tasks including (a) quality assessment of conditional generative models, (b) posterior accuracy assessment in Bayesian inference, and (c) Bayesian model comparison.

## 2. Preliminaries

In this section, we introduce the notation, define the problem we aim to solve, and describe the key concepts underlying our method.

### 2.1. Problem Statement and Notations

We assume a probability space $(\Omega, \mathcal{F}, \mathbb{P})$. Let $X$ and $Y$ be random variables defined as measurable functions $X : \Omega \to \mathcal{X}$ and $Y : \Omega \to \mathcal{Y}$, where the codomains $\mathcal{X} \subseteq \mathbb{R}^{d_x}$ and $\mathcal{Y} \subseteq \mathbb{R}^{d_y}$ are equipped with the Lebesgue measure. We consider a setting where we have access to a true probabilistic model $\mathcal{M}^*$, from which we can sample $L$ joint pairs $\{(x_i^*, y_i^*)\}_{i=1}^{L} \sim p(x, y \mid \mathcal{M}^*)$.

Our goal is to quantify the fidelity of a candidate conditional distribution $p(y \mid x, \mathcal{M})$ against the true, unknown one $p(y \mid x, \mathcal{M}^*)$ for all $x \in \{x_i^*\}_{i=1}^{L}$. We assume we can draw $N$ i.i.d. samples $\{y_j\}_{j=1}^{N} \sim p(y \mid x_i^*, \mathcal{M})$ per observations $x_i^*$, but we have access to only a single realization $y_i^*$ from the true conditional distribution per observations $x_i^*$ as we only have access to joint pairs $(x_i^*, y_i^*)$.

The candidate model $\mathcal{M}$ may represent either a machine learning model or a sampling method, when the goal is to validate the conditional distribution produced by it, or a candidate model $(x, y) \sim p(x, y \mid \mathcal{M})$ (e.g. a numerical simulator of a physical system) when the goal is to test whether its joint distribution is consistent with the true data-generating process. This framework also applies to non-conditional distributions.

### 2.2. PQMass

PQMass (Lemos et al., 2025) is a sample-based method designed to assess whether two sets of samples originate from the same distribution.

Formally, two distributions $p(y \mid \mathcal{M}^*)$ and $p(y \mid \mathcal{M})$ are equal if their probability measures are the same over all measurable sets $\mathcal{R} \subseteq \Omega$: $\mathbb{P}_{\mathcal{M}^*}(\mathcal{R}) = \mathbb{P}_{\mathcal{M}}(\mathcal{R}) \quad \forall \mathcal{R} \subseteq \Omega$. The probability measures of a region $\mathcal{R}$ under model $\mathcal{M}$ can be unbiasedly estimated as:

$$\mathbb{P}_{\mathcal{M}}(\mathcal{R}) = \int_{\mathcal{R}} p(y \mid \mathcal{M}) dy \approx \frac{1}{N} \sum_{i=1}^{N} \mathbb{1}(y_i \in \mathcal{R}), \quad (1)$$

where $\{y_i\}_{i=1}^{N} \sim p(y \mid \mathcal{M})$ are $N$ independent samples. Furthermore, the number of samples falling within $\mathcal{R}$, $n = \sum_{i=1}^{N} \mathbb{1}(y_i \in \mathcal{R})$, follows a binomial distribution:

$$n \sim \mathcal{B}\left(N, \mathbb{P}_{\mathcal{M}}(\mathcal{R})\right). \quad (2)$$

This property allows for the comparison of two distributions by comparing the binomial distributions over multiple chosen regions $\mathcal{R}$. Recognizing that the counts of samples in any collection of disjoint regions follow a multinomial distribution, PQMass actually compares the multinomial distributions induced by model $\mathcal{M}$ and model $\mathcal{M}^*$ using the Pearson $\chi^2$ goodness-of-fit test (Rao, 1948).

We note that the limitations of PQMass are significant for our problem. First, it is designed for comparing marginal distributions $p(y \mid \mathcal{M})$ and cannot evaluate the quality of a conditional distribution $p(y \mid x, \mathcal{M})$ for all observations $x$. Second, it is a frequentist approach where the resulting p-value is inherently sensitive to the number of samples. As the number of samples increases, the p-value decreases due to increased statistical power, even if the absolute deviation between distributions remains constant. This sample sensitivity makes the PQMass p-value ill-suited for the purpose of model comparison, where a stable, scalar fidelity

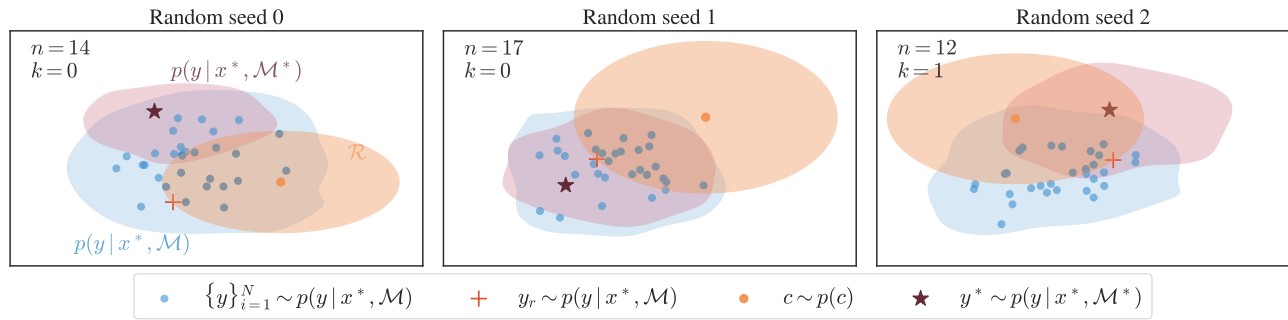

*Figure 1.* 2D illustration of the quantities used in Mira to compare two conditional distributions. Following PQMass (Lemos et al., 2025), we compare the number of samples from each distribution falling in a region $\mathcal{R}$ (orange). Each region is defined by a random center $c$ (orange point) sampled from $p(c)$, and a radius set by the distance to a random candidate sample $y_r$ (orange cross). A single sample $y^*$ (red star) is drawn from the true conditional $p(y \mid x^*, \mathcal{M}^*)$ (red), and compared to $N$ samples $\{y\}_{i=1}^{N}$ (blue dots) from the candidate $p(y \mid x^*, \mathcal{M})$ (blue). Three random scenarios are shown.

score is necessary. Finally, the PQMass method cannot be adapted to our setting because it relies on having multiple samples from both distributions being compared, making it incompatible with the single ground-truth sample constraint ($y^*$) that defines our problem. This justifies shifting from a frequentist approach to a Bayesian one.

## 3. Mira

### 3.1. Proposal

We propose to compare conditional distributions based on the principle that they are identical if they assign equal mass to all measurable sets for all conditioning variables. Following the PQMass framework, we approximate the mass by comparing sample counts within randomly constructed regions. In Mira, we define each region $\mathcal{R}$ as

$$\mathcal{R} = \{y \in \mathcal{Y} \mid d(y, c) \leq d(y_r, c)\}, \quad (3)$$

with $d$ a distance metric, $y_r \sim p(y \mid x^*, \mathcal{M})$, and $c \sim p(c)$ where $p(c)$ is any distribution[1]. We exclude the case where $c = y_r$, that is, where $\mathcal{R}$ is a set of measure 0. Additionally, we denote

$$n = \sum_{j=1}^{N} \mathbb{1}(y_j \in \mathcal{R}), \quad \text{and} \quad k = \mathbb{1}(y^* \in \mathcal{R}), \quad (4)$$

where $n$ is the number of candidate draws falling in $\mathcal{R}$, and $k$ indicates whether $y^*$ lies in $\mathcal{R}$. These two random variables follow Binomial distributions

$$n \sim \mathcal{B}(N, \lambda_n), \quad \text{and} \quad k \sim \mathcal{B}(1, \lambda_k),$$

with $\lambda_n$ and $\lambda_k$ denoting respectively the mass of $p(y \mid x^*, \mathcal{M})$ and $p(y \mid x^*, \mathcal{M}^*)$ in $\mathcal{R}$, that is

$$\lambda_n = \int_{\mathcal{R}} p(y \mid x^*, \mathcal{M}) dy , \quad \lambda_k = \int_{\mathcal{R}} p(y \mid x^*, \mathcal{M}^*) dy.$$

---

[1]In Appendix E.5, we empirically investigate the sensitivity of Mira to the choice of distribution and support for $p(c)$.

We provide in Figure 1 an illustration of our framework.

Because we observe only a single realization $y^*$ from the true conditional distribution for each $x^*$, the corresponding probability mass $\lambda_k \mid x^*, y_r, c$ cannot be estimated. As a result, a direct frequentist comparison of $\lambda_n \mid x^*, y_r, c$ and $\lambda_k \mid x^*, y_r, c$, as performed in PQMass, is not feasible, since it requires multiple samples from both distributions. To address this limitation, we adopt a Bayesian perspective and introduce the *Mira statistic* $p(k \mid n)$, defined as the probability that the true sample $y^*$ lies within a region $\mathcal{R}$, given that $n$ samples drawn from the candidate conditional distribution fall within the same region. This probability is defined under the null hypothesis that the candidate and true conditional distributions are identical. In the following section, we show that, with our randomized construction of $\mathcal{R}$, the Mira statistic admits a closed-form expression.

### 3.2. Deriving the Mira statistic

The random construction of the region $\mathcal{R}$ where the reference point $y_r$ is drawn from the candidate distribution, guarantees a key property of the mass $\lambda_n$:

**Proposition 3.1** (Uniform Mass). *Let $\mathcal{R}$ be the random region defined in Equation 3. For any realization of the true observation $x^*$ and any realization of the center $c$, the probability mass $\lambda_n$ assigned to the resulting region $\mathcal{R}$ follows a Uniform distribution on $[0, 1]$, that is, $\lambda_n \mid x^*, c \sim \mathcal{U}[0, 1]$. As a consequence, the marginal distribution of $\lambda_n$ is also Uniform on $[0, 1]$.*

*Proof.* See Appendix A.1 □

This result enables a closed-form derivation of the Mira statistic under the null hypothesis that the candidate and true conditional distributions coincide, as stated in the proposition below.

**Proposition 3.2** (Mira statistic). *Assume the null hypothesis holds, such that the deterministic variables $\lambda_n \mid x^*, y_r, c$ and $\lambda_k \mid x^*, y_r, c$ are equal almost surely. Under this assumption, the probability of $k$ given that $n$ samples from $\{y\}_{j=1}^N$ fall in $\mathcal{R}$, is given by Laplace's rule of succession:*

$$p(k \mid n) = \frac{n+1}{N+2}\mathbb{1}(k=1) + \frac{N-n+1}{N+2}\mathbb{1}(k=0). \quad (5)$$

*Proof.* See Appendix A.2 □

We note that the expression of the Mira statistic is marginalized over all regions $\mathcal{R}$ and observations $x^*$ (see Appendix A.2). To obtain the accuracy for a candidate model $\mathcal{M}$, one needs to evaluate the Mira statistic for all $(k, n) \sim p(k, n)$. Hence, we define the *Mira score* as the expectation of the Mira statistic over all sources of randomness $Z = (\mathcal{R}, y^*, x^*, \{y\}_{i=1}^N)$:

$$\mu_{\text{Mira}}(\mathcal{M}) = \mathbb{E}_{p(Z)}\left[\mathbb{E}_{p(k,n|Z)}\left[p(k \mid n)\right]\right]. \quad (6)$$

### 3.3. Deriving Theoretical Quantities

The Mira statistic $p(k \mid n)$ is a random variable, as it is a function of the random counts $n$ and $k$. The following proposition characterizes its asymptotic distribution under the null hypothesis.

**Proposition 3.3** (Mira statistic law). *Suppose the two deterministic variables $\lambda_n \mid x^*, y_r, c$ and $\lambda_k \mid x^*, y_r, c$ are equal almost surely. Then the random variable $P_N := p(k \mid n)$ converges in distribution to a Beta$(2, 1)$ random variable as $N$ goes to infinity:*

$$P_N \xrightarrow{d} Beta(2, 1). \quad (7)$$

*Proof.* See Appendix A.3 □

In particular, its mean and variance converge to those of the Beta$(2, 1)$ distribution.

**Corollary 3.4** (Mira statistic moments). *Suppose that $\lambda_n \mid x^*, y_r, c = \lambda_k \mid x^*, y_r, c$ almost surely. Then, the mean and variance of the random variable $P_N$ bounded between $0$ and $1$, converge to the moments of the Beta(2,1) distribution, that is,*

$$\mathbb{E}_{p(k,n)}[P_N] \to \tfrac{2}{3} \quad as\ N \to \infty \quad (8)$$
$$Var_{p(k,n)}[P_N] \to \tfrac{1}{18} \quad as\ N \to \infty. \quad (9)$$

*Proof.* See Appendix A.4 □

This result provides a theoretical mean to which to compare the empirical score defined in Equation 6. Additionally, the theoretical variance can be used to quantify the discrepancy

between the empirical score and the true score, normalized by the theoretical uncertainty under perfect calibration.

We note that while this is a necessary condition for calibration, Mira is not a sufficient condition. In Appendix B, we show that the Mira score can be modified to render it a sufficient condition, at the cost of sample efficiency.

Finally, leveraging the closed-form expression of the Mira statistic (Proposition 3.2) combined with Proposition 3.1, one can derive additional theoretical quantities. For instance, determining the lower bound of the Mira score:

**Proposition 3.5** (Lower bound on the Mira score). *For any candidate model $\mathcal{M}$,*

$$\mu_{\text{Mira}}(\mathcal{M}) \geq \frac{1}{2}, \quad (10)$$

*with equality approached if the candidate and true distributions have disjoint supports.*

*Proof.* See Appendix A.5 □

### 3.4. Algorithm

In practice, the Mira score is approximated using Monte Carlo (MC) integration as

$$\mu_{\text{Mira}}(\mathcal{M}) \approx \frac{1}{L \cdot L_r} \sum_{l,j} p(k_{l,j} \mid n_{l,j}), \quad (11)$$

with $L$ the number of fiducials $(x^*, y^*)$ from the true model $p(x, y \mid \mathcal{M}^*)$ and $L_r$ the number of regions. Below, we provide the pseudo-code to compute the Mira score.

---
**Algorithm 1** Computing Mira score for model $\mathcal{M}$

---
1: **Input:** Number of fiducial $L$, region $L_r$, and samples $N$. Distribution $p(c)$ and metric $d$. Forward model $p(x, y \mid \mathcal{M}^*)$ and candidate distribution $p(y \mid x, \mathcal{M})$.
2: Initialize `score` $\leftarrow 0$
3: **for** $i = 1$ to $L$ **do**
4:     Draw $x^*, y^* \sim p(x, y \mid \mathcal{M}^*)$
5:     Draw $\{y_j\}_{j=1}^N \sim p(y \mid x^*, \mathcal{M})$
6:     **for** $\ell = 1$ to $L_r$ **do**
7:         Draw $c \sim p(c),\ y_r \sim p(y \mid x^*, \mathcal{M})$
8:         $n \leftarrow \sum_j \mathbb{1}[d(y_j, c) \leq d(c, y_r)]$
9:         $k \leftarrow \mathbb{1}[d(c, y^*) \leq d(c, y_r)]$
10:        `score` $+= \frac{n+1}{N+2}$ if $k = 1$, else $+= \frac{N-n+1}{N+2}$
11:    **end for**
12: **end for**
13: **return** $\mu_{\text{Mira}}(\mathcal{M}) = \frac{\text{score}}{L \cdot L_r}$

---

We estimate the uncertainty of the Mira score using nonparametric bootstrapping over the fiducial set. Let $\mathcal{D} = \{(x_k^*, y_k^*)\}_{k=1}^L$ denote the set of $L$ fiducials. To estimate

the variance, we perform $B$ bootstrap iterations. In each iteration $b$, we construct a resampled dataset $\mathcal{D}_b$ by drawing $L$ indices from $\{1, \ldots, L\}$ with replacement. We then compute the aggregate Mira score $\mu_b$ for this resampled set using Algorithm 1. The uncertainty we report is the variance of the bootstrap means. This approach captures the variability in the finite set of fiducials and accounts for the Monte Carlo integration variability over regions.

We note that Mira plays a pivotal role in alleviating the curse of dimensionality, as it transforms the comparison of high-dimensional distributions into a comparison of one-dimensional probability mass functions. By defining regions in a data-dependent manner, Mira is able to identify mis-specified conditional distributions even when only relatively small sample sizes are available, as demonstrated by the sensitivity analysis presented in Appendix E.5.

## 4. Related Work

A number of methods have been proposed to assess the accuracy of conditional distributions $p(y \mid x)$, especially in settings where access to the true distribution is limited to samples from the joint $p(x, y)$. We briefly review key approaches that Mira complements or improves upon.

**Coverage Probability Tests** A cornerstone of model validation is coverage probability, which measures how often a region or interval derived from a conditional distribution $\hat{p}(y \mid x)$ contains the true outcome $y$, across repeated draws $(x, y) \sim p(x, y)$ (Neyman, 1937; Talts et al., 2018). While widely used, how discriminative these tests are depends on the choice of credible region, and their result does not quantify the degree of bias measured when they reveal mis-specification (Patel et al., 2024).

A common method for constructing these regions is to use High Posterior Density (HPD) intervals, which define the smallest region of $p(y \mid x)$ containing a specified probability mass (Box & Tiao, 1973). These intervals capture the most probable outcomes and have gained popularity as an accuracy metric in Simulation-Based Inference approaches (Hermans et al., 2022). However, their construction requires access to a density function, which is often unavailable, and their coverage can be ambiguous in high-dimensional or multimodal settings (Betancourt, 2018; Hyndman, 1996). Moreover, achieving nominal coverage with this choice of region is not a sufficient condition for accuracy.

Recently, methods have emerged to construct guaranteed coverage regions in likelihood-free settings. For instance, TARP (Lemos et al., 2023) was proposed as a randomized scheme to construct regions in a way that can guarantee accuracy even for high-dimensional and multimodal conditional distributions. WALDO (Masserano et al., 2023), on the other hand, constructs valid frequentist confidence re-

gions for model parameters. See also related developments in frequentist-calibrated posterior inference (Dey, 2022; Dey et al., 2025). However, the problems of quantifying degrees of accuracy across multiple models and of interpretability in large dimensional spaces remain unaddressed by these tests.

**Probability Integral Transform Tests** Beyond testing specific regions, some diagnostic tools focus on the entire distribution, like the Probability Integral Transform (PIT) (Dawid, 1984; Hyndman, 1996; Anderson, 1996; Diebold et al., 1998). For a continuous univariate variable $y$, the PIT is the value of the cumulative distribution function (CDF) evaluated at the true observation, $z = F_{y|x}(y \mid x)$. If the proposal distribution is perfectly calibrated, these $z$ values must be uniformly distributed, with deviations revealing systematic biases (Gneiting et al., 2007). The agreement between the $z$ and uniform distributions is often tested with a p-value or Kolmogorov-Smirnov (KS) test, making this class of test naturally framed in a frequentist framework. However, when the Agnesti-Coull adjustment is applied to PIT, the empirical CDF can be smoothed with a different choice or prior, reframing this class of test in a Bayesian context. While there exists a connection which is made obvious through the rank-ordering reformulation of Mira in Appendix B.1, generalizing PIT diagnoses to high-dimensional outputs is non-trivial. Common approaches therefore rely on extensions such as dimensionality reduction (Gneiting et al., 2008), the Rosenblatt transformation (Rosenblatt, 1952), or, more practically in modern sample-based settings, on rank-based diagnostics as used in Simulation-Based Calibration (SBC) (Gneiting et al., 2008; Talts et al., 2018). However, SBC evaluates one dimension at a time, which ignores joint correlation and makes it difficult to scale to high-dimensional setting, both computationally and in terms of interpretability. In addition, SBC typically requires a large number of fiducial samples $(x^*, y^*) \sim p(x, y)$ (i.e., a large number of conditionals $p(y \mid x = x^*)$) to obtain reliable rank histograms.

**Predictive Tests** Predictive tests compare model-generated data to observed data. Prior Predictive Checks involve generating data from the model using only the prior distribution, $y_{\text{pred}} \sim p(y)$, $x_{\text{pred}} \sim p(x \mid y_{\text{pred}})$, exhibiting the implications of the chosen priors and model structure (Box, 1980). Posterior Predictive Checks (PPCs), on the other hand, evaluate model adequacy after fitting (Rubin, 1984; Meng, 1994; Gelman et al., 1996). They operate by comparing observed data to replicated data drawn from the posterior predictive distribution, $x_{\text{rep}} \sim p(x \mid y)$, $y \sim \hat{p}(y \mid x_{\text{obs}})$. While widely used, PPCs operate in observation space and may miss errors in the posterior $\hat{p}(y \mid x)$, especially when the mapping $y \to x$ is complex or non-injective. A key challenge in PPCs is selecting and interpreting discrepancy statistics. Evidence-based methods offer solutions to some of these problems. For instance, Neural Evidence Estima-

tors have been proposed to directly and efficiently compute the Bayesian evidence (or marginal likelihood), enabling model comparison, particularly for models with intractable likelihoods (Radev et al., 2023; Jeffrey & Wandelt, 2024).

**Two-Sample Tests** Standard metrics like Maximum Mean Discrepancy (MMD) (Gretton et al., 2012), Wasserstein distance (Villani, 2009), and Classifier Two-Sample Tests (C2ST) (Lopez-Paz & Oquab, 2017) provide powerful methodology for assessing differences between probability distributions. However, their application typically presupposes access to multiple i.i.d. samples from each distribution under comparison. In our setting, we seek to evaluate a candidate conditional distribution $\hat{p}(y \mid x)$ against the true but unknown conditional $p(y \mid x)$, for which, in practice, only a single ground-truth realization $y$ is observed for each $x$, as data are available only in the form of joint samples $(x, y) \sim p(x, y)$. A recent variant of C2ST designed to operate under such constraints has been proposed by Linhart et al. (2023). While this approach is promising, it inherits from classical C2ST the reliance on training a classifier to discriminate between two sets of samples, and therefore requires a substantial number of samples. Moreover, its scalability to high-dimensional settings is unclear.

## 5. Experiments

In Section 5.1, we show that Mira detects overconfident, underconfident, and biased posteriors, and in Section 5.2 that it identifies inaccurate conditional distributions. We then move to more realistic Bayesian inference problems, demonstrating in Section 5.3.1 that Mira detects likelihood misspecification, and in Section 5.3.2 that it identifies misspecified priors and incorrect noise models, enabling Bayesian model comparison of different physical models. In Appendix E.1.1 and Appendix E.1.2, we further evaluate Mira on inverse imaging problems, comparing different generative models in their ability to approximate posterior distributions. Notably, the experiments in Section 5.3.2, Appendix E.1.1, and Appendix E.1.2 involve complex, high-dimensional data. In addition, Appendix E.5 presents a sensitivity analysis demonstrating the robustness of Mira to key design choices, including dimensionality, the number of regions, the numbers of conditional and true samples ($N$ and $L$), the distance metric $d$ used to define regions, and the distribution $p(c)$ used to sample region centers. In Appendix E.4, we show how Mira can detect uninformative posterior estimator. Finally, in Appendix E.2, we show strong agreement between Mira and the Bayes factor, and in Appendix E.3 we compare Mira to existing conditional accuracy scores, highlighting failure modes of competing methods that Mira avoids.

To confirm Mira outputs, we use the TARP (Lemos et al., 2023) Expected Coverage Probability (ECP) test that oper-

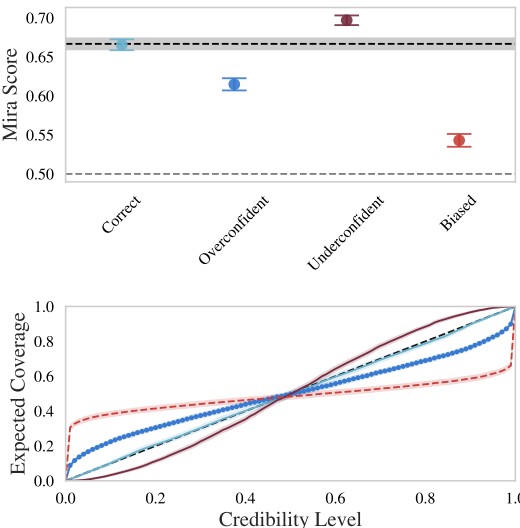

*Figure 2.* Test of Mira's ability to detect overconfident, underconfident, and biased distributions. **Top:** The Mira score separates these cases, with the true distribution attaining the expected value of $2/3$. The shaded gray region indicates the theoretical estimation uncertainty, $2/3 \pm \sqrt{1/18L}$. **Bottom:** TARP confirms Mira's results, with the correct distribution lying on the diagonal.

ates in the same joint samples setting. The ECP assesses, for all conditioning variables, the frequency with which conditional distribution samples fall within a credible region. In contrast to Highest Posterior Density (HPD) coverage, TARP provides a procedure to compute this ECP such that the conditional distribution is considered accurate if and only if, for every credible region with credibility $1 - \alpha$, the corresponding ECP equals $1 - \alpha$. Accordingly, in the associated plots, an accurate conditional distribution yields a curve that coincides with the diagonal line. A key limitation of TARP is that deviations from the diagonal line are difficult to quantify, making direct model comparison challenging. This comparison thus also enables us to emphasize concrete cases where Mira is of particular importance compared to TARP (e.g., the blue, red, and maroon curves in Figure 4).

In the following, samples $\{y_j\}_{j=1}^N$ are normalized to $[0, 1]$ with $y^*$ following the same normalization, $p(c) = \mathcal{U}(0, 1)$, we use the euclidean distance to build the region $\mathcal{R}$, and we generate 100 regions $\mathcal{R}$ per true sample $(x^*, y^*)$.

### 5.1. Detecting Over and Under Confidence

To evaluate Mira on detecting overconfident, underconfident, and biased conditional distributions, we replicate the toy inference setup of Lemos et al. (2023). We sample $L = 1\,000$ parameters $\theta^* \sim \mathcal{U}([-5, 5]^2)$ and, for each, define a covariance $\Sigma$ that is diagonal with $\log \sigma_i \sim \mathcal{U}(-5, -1)$. The candidate distribution is fixed to $\hat{p}(y \mid \theta^*) = \mathcal{N}(\theta^*, \Sigma)$ across all four scenarios. What varies is how the ground-truth sample $y^*$ is drawn. Specifically, we consider the

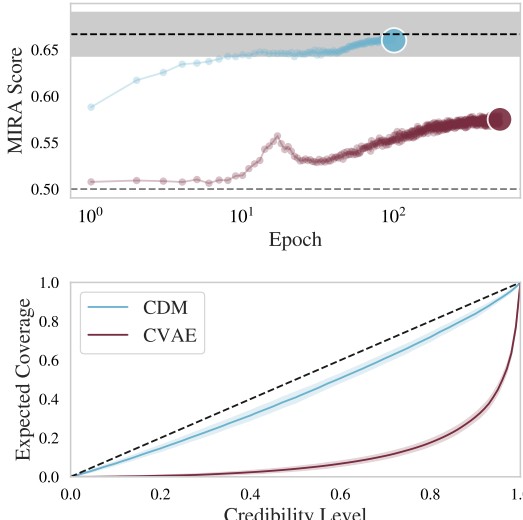

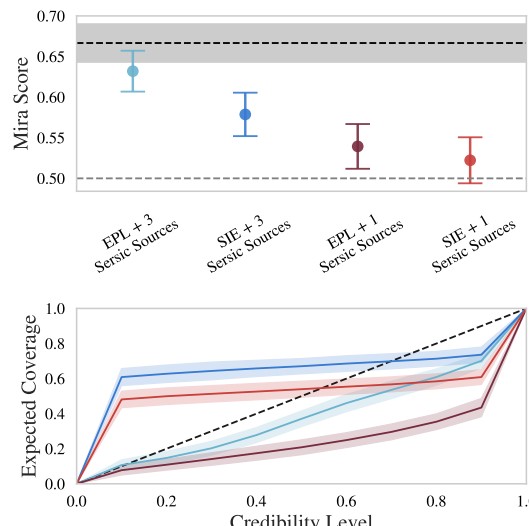

*Figure 3.* Comparison of two conditional generative models learning $p(\text{MNIST image} \mid \text{label})$. **Top:** Mira score as a function of training (100 epochs for the Conditional Diffusion Model, CDM, 500 epochs for Conditional VAE). The CDM approaches the ideal score of $2/3$, while the conditional VAE performs worse. The gray region indicates the theoretical uncertainty, $2/3 \pm \sqrt{1/18L}$. **Bottom:** TARP validates the results.

correct model to be $p(y) = \mathcal{N}(\theta^*, \Sigma)$, the overconfident model is $p(y) = \mathcal{N}(\theta^*, 3\Sigma)$, the underconfident is $p(y) = \mathcal{N}(\theta^*, 0.5\Sigma)$, and the biased use $p(y) = \mathcal{N}(\theta^*, \Sigma)$ but shift the candidate as $\mathcal{N}(\theta^* - \text{sign}(\theta^*) \, Z(1 - |\theta^*|/5) \, \sigma, \Sigma)$, with $Z$ the inverse survival function. To compute Mira, we draw $N = 500$ candidate samples per observation. As shown in Figure 2, Mira correctly identifies the true model, assigns lower scores to the overconfident and biased cases, and a higher one to the underconfident case. These findings are consistent with the TARP coverage curves. Beyond detecting these failure modes, the sign of the deviation from $2/3$ is itself informative: scores below $2/3$ are consistent with overconfidence, while scores above $2/3$, underconfidence (see Appendix C).

## 5.2. Conditional Distribution

We demonstrate that Mira can be employed to detect inaccurate conditional distribution. To this end, we consider the MNIST digits dataset (LeCun et al., 1998) and use Mira to assess how accurately a Conditional Variational Autoencoder (CVAE, Ivanov et al., 2019; Sohn et al., 2015) and a conditional diffusion model (Ho et al., 2020) approximate the true conditional distribution $p(\text{image} \mid \text{label})$. Architecture details are provided in Appendix D.1. For each generative model, we draw $N = 1\,000$ samples per label and compare them with 100 randomly selected MNIST test images, which yields $L = 100$ true samples for the MC approximation (see Equation 11). As shown in Figure 3, the conditional diffusion model exhibits superior performance relative to the CVAE, attaining a Mira score that is close to

*Figure 4.* Comparison of strong gravitational lensing likelihood models. **Top:** Mira scores identify the correct model and rank them accurately. The gray band is the theoretical uncertainty, $2/3 \pm \sqrt{1/18L}$. **Bottom:** TARP coverage plots. While the correctly specified model can be identified, the curves of the remaining models are difficult to interpret and rank.

the expected theoretical mean of $2/3$, whereas the CVAE attains a score of approximately $0.56$. These findings are corroborated using TARP. In Figure 6, we present generated samples from both models for each label, confirming that the conditional diffusion model produces images of higher perceptual fidelity than those generated by the CVAE.

## 5.3. Bayesian Inference and Bayesian Model Comparison

Bayesian inference aims to infer the posterior distribution $p(y \mid x, \mathcal{M}^*) \propto p(x \mid y, \mathcal{M}^*)p(y \mid \mathcal{M}^*)$ of a forward model $\mathcal{M}^*$ of parameters $y$ and observations $x$. In this context, Mira can provide a computationally efficient approach to Bayesian model comparison, particularly in high-dimensional settings. By quantifying the discrepancy between candidate and reference posteriors, Mira directly ranks models in parameter space without requiring often intractable evidence computations. It evaluates model fidelity based on parameter distributions rather than data likelihoods, avoiding the traditional bottlenecks of the Bayes factor while offering complementary insights (see Appendix E.2).

### 5.3.1. DETECTING PHYSICAL MODEL MISMATCH

We aim to demonstrate that Mira can be used to detect likelihood mismatch. For this, we consider an astrophysical inference problem. Strong gravitational lensing is the distortion of background galaxy images (sources) by intervening matter (the lens). Our inference task is to jointly infer the lens and source parameters from a distorted image. The true model $\mathcal{M}^*$ is a forward model where the distorted image

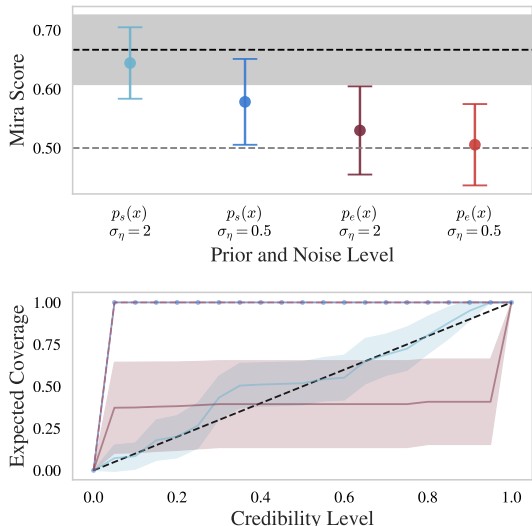

*Figure 5.* Comparison of prior and noise models for source reconstruction in strong gravitational lensing. **Top:** Mira scores identify the correct model and accurately rank the remaining candidates, despite the limited data. The shaded gray band indicates the theoretical uncertainty, $2/3 \pm \sqrt{1/18L}$. **Bottom:** TARP coverage plots. Although the correctly specified model can be identified, the remaining models are difficult to reliably rank.

is generated with an elliptical power-law (EPL) lens and 3 Sérsic sources (see Appendix D.2 for details). We compare four candidate forward models that differ in lens type and source count: EPL + 3 Sérsic sources (ground truth), singular isothermal ellipsoid (SIE) + 3 Sérsic sources, EPL + 1 Sérsic source, and SIE + 1 Sérsic source. All models share the same prior distribution over parameters. To run Mira, we draw $L = 100$ true samples from the true forward model and use Metropolis-adjusted Langevin sampling (MALA; Roberts & Tweedie, 1996) to obtain 100 posterior distributions, each with $N = 20\,000$ samples in a 13-dimensional space (see Figure 7 for the samples). Figure 4 shows that Mira correctly identifies EPL + 3 Sérsic sources as the true model, yielding a score closest to $2/3$. As expected, the results indicate that matching the number of sources is more critical than matching the lens profile. While TARP is initially used to validate our results, this example highlights the importance of a score such as Mira, which provides an unambiguous model ranking in settings where TARP curves are difficult to interpret.

### 5.3.2. DETECTING PRIOR DISTRIBUTION AND NOISE MODEL SHIFTS

This experiment tests whether Mira can detect wrong prior distribution and likelihood noise, and whether it performs well in high dimensions with few true samples. We consider a high-dimensional strong gravitational lensing inference task: inferring the source galaxy image from a noisy lensed image. In our forward models, the prior generates a pixelated galaxy image of size $64 \times 64 \times 3$, which is then

lensed using a fixed forward model. Gaussian noise with standard deviation $\sigma_{\boldsymbol{\eta}}$ is added. Within this setup, we define four simulation models by varying the prior and noise: (1) spiral galaxies $p_s(x)$ and $\sigma_{\boldsymbol{\eta}} = 2$ (ground truth), (2) spiral galaxies $p_s(x)$ and $\sigma_{\boldsymbol{\eta}} = 0.5$, (3) elliptical galaxies $p_e(x)$ and $\sigma_{\boldsymbol{\eta}} = 2$, (4) elliptical galaxies $p_e(x)$ and $\sigma_{\boldsymbol{\eta}} = 0.5$. We draw $L = 16$ true samples and, for each, generate $N = 64$ posterior samples using a score-based diffusion model (see Appendix D.3 for details). Example observations (lensed galaxies + noise), parameters (source galaxies), and posterior samples (inferred sources) for each configuration are shown in Appendix D.3. We then apply Mira to evaluate the four models. As shown in Figure 5, Mira ranks them correctly: the true model is closest to $2/3$; and having the true prior distribution is more critical than having the right noise model. This demonstrates that Mira scales to high dimensionality ($3 \times 64 \times 64$) and can detect wrong prior and noise model, even with few true samples $L$ (see Appendix E.5 for further investigation of this regime).

## 6. Discussions and Conclusion

We introduced Mira, a method to assess the accuracy of candidate conditional distributions using only joint samples from the true data generating process. By assigning a score to each candidate, Mira naturally enables model comparison. It is particularly useful for Bayesian model comparison as a complement to the Bayes factor, which operates in observation space, since Mira instead ranks models by comparing their posteriors in parameter space.

Mira relies on the principle that two distributions coincide if they assign equal probability mass to all regions. By constructing regions from samples of the candidate conditional distribution, under the null hypothesis, we obtain a closed-form expression for the probability that a true sample falls within a region given the number of candidate samples in that region. Averaging this quantity across conditioning variables and regions yields the Mira score. The analytic formulation further allows us to derive theoretical reference value ($2/3$) and theoretical uncertainty ($1/18$) for correctly specified models. Unlike many existing approaches, Mira scales naturally to high-dimensions, does not require neural network training, and operates using only joint samples.

Across a broad set of experiments, Mira consistently identifies correctly specified models and reliably ranks misspecified alternatives. Sensitivity studies in Appendix E.5 show that the method remains robust to dimensionality and to the number of true and conditional samples. We also demonstrate performance on "realistic" high-dimensional inference tasks with limited samples (e.g., Section 5.3.2). Finally, comparisons with existing conditional accuracy diagnostics in Appendix E.3 reveal scenarios in which the tested methods fail while Mira remains reliable.

## Impact Statement

This paper presents work whose goal is to advance the field of machine learning. There are many potential societal consequences of our work, none of which we feel must be specifically highlighted here.

## Acknowledgment

The authors would like to thank Benjamin Remy for the valuable discussions. This work is supported by the Simons Collaboration on 'Learning the Universe' and by Schmidt Sciences, a philanthropic initiative founded by Eric and Wendy Schmidt as part of the Virtual Institute for Astrophysics (VIA). This research work is partially supported by European Commission funding under the Marie Skłodowska-Curie grant agreement No. 101127936 - DeMythif.AI. This work is partially supported by France 2030 funding, managed by the National Research Agency (ANR), as part of IA CLUSTER program, reference ANR-23-IACL-0003 - DATAIA CLUSTER. This work was partially supported by the Fond de Recherche du Quebec through grant doi.org/10.69777/348060, and by computational resources provided by Calcul Quebec and the Digital Research Alliance of Canada. Y.H. and L.P. acknowledge support from the Canada Research Chairs Program, the National Sciences and Engineering Council of Canada through grants RGPIN-2020-05073 and 05102.

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

# A. Proofs

In this section, we provide the proofs of the Propositions and Corollary of the paper.

## A.1. Uniform Mass

**Proposition A.1** (Uniform Mass). *Let $\mathcal{R}$ be the random region defined in [Equation 3](). For any realization of the true observation $x^*$ and any realization of the center $c$, the probability mass $\lambda_n$ assigned to the resulting region $\mathcal{R}$ follows a Uniform distribution on $[0,1]$, that is, $\lambda_n \mid x^*, c \sim \mathcal{U}[0,1]$. As a consequence, the marginal distribution of $\lambda_n$ is also Uniform on $[0,1]$.*

*Proof.* We prove using the probability integral transform (PIT) that $p(\lambda_n) = \mathcal{U}[0,1]$ where $\lambda_n$ is the random variable defined as

$$\lambda_n = \int p(y \mid x^*, \mathcal{M})\mathbb{1}\left(d(y,c) \leq d(y_r,c)\right) dy, \tag{12}$$

with $x^* \sim p(x \mid y^*, \mathcal{M}^*)$, $y_r \sim p(y \mid x^*, \mathcal{M})$ and $c \sim p(c)$. We start by considering the conditioned random variable $\lambda \mid x^*, c$ and derive its cumulative distribution function

$$
\begin{aligned}
F_{\lambda_n \mid x^*, c}(\alpha) &= \int p(\lambda_n \mid x^*, c)\mathbb{1}(\lambda_n \leq \alpha)d\lambda_n \\
&= \int \int p(\lambda_n \mid x^*, c, y_r)p(y_r \mid x^*, \mathcal{M})\mathbb{1}(\lambda_n \leq \alpha)d\lambda_n dy_r \\
&= \int p(y_r \mid x^*, \mathcal{M})\mathbb{1}\left(\left(\int p(y \mid x^*, \mathcal{M})\mathbb{1}\left(d(y,c) \leq d(y_r,c)\right) dy\right) \leq \alpha\right) dy_r,
\end{aligned} \tag{13}
$$

Where we used that $p(\lambda_n \mid x^*, c, y_r)$ is a Dirac. Recognizing that $\lambda_n \mid x^*, c = F_D\left(d(y_r, c)\right)$ where $D$ is the continuous random variable $d(y,c)$ with $y \sim p(y \mid x^*, \mathcal{M})$, we derive

$$
\begin{aligned}
F_{\lambda_n \mid x^*, c}(\alpha) &= \int p(y_r \mid x^*, \mathcal{M})\mathbb{1}\left(F_D\left(d(y_r, c)\right) \leq \alpha\right) dy_r \\
&= \int p(y_r \mid x^*, \mathcal{M})\mathbb{1}\left(d(y_r, c) \leq F_D^{-1}(\alpha)\right) dy_r \\
&= F_D\left(F_D^{-1}(\alpha)\right) \\
&= \alpha.
\end{aligned} \tag{14}
$$

Hence, $\lambda_n \mid x^*, c \sim \mathcal{U}[0,1]$ for all $x^*, c \sim p(x \mid y^*, \mathcal{M}^*)p(c)$. Thus the marginal $p(\lambda_n) = \mathcal{U}[0,1]$. $\square$

## A.2. Mira Statistic Derivation

**Proposition A.2** (Mira statistic). *Assume the null hypothesis holds, such that the deterministic variables $\lambda_n \mid x^*, y_r, c$ and $\lambda_k \mid x^*, y_r, c$ are equal almost surely. Under this assumption, the probability of $k$ given that $n$ samples from $\{y\}_{j=1}^N$ fall in $\mathcal{R}$, is given by Laplace's rule of succession:*

$$p(k \mid n) = \frac{n+1}{N+2}\mathbb{1}(k=1) + \frac{N-n+1}{N+2}\mathbb{1}(k=0). \tag{15}$$

*Proof.* We begin by marginalizing over the region $\mathcal{R}$ (i.e., $y_r$ and $c$), the observation $x^*$ and the shared deterministic parameter $\lambda \mid x^*, \mathcal{R}$, where we defined $\lambda \mid x^*, \mathcal{R} = \lambda_n \mid x^*, \mathcal{R} = \lambda_k \mid x^*, \mathcal{R}$ under the null hypothesis:

$$
\begin{aligned}
p(k \mid n) &= \int \int \int p(k, \lambda, \mathcal{R}, x^* \mid n)d\lambda d\mathcal{R}dx^*, \\
&= \int \int \int \frac{p(k, n \mid \lambda, \mathcal{R}, x^*)p(\lambda, \mathcal{R}, x^*)}{p(n)}d\lambda d\mathcal{R}dx^*,
\end{aligned}
$$

Because $k$ and $n$ are independent of $\mathcal{R}$ and $x^*$ given $\lambda$ we have

$$
\begin{aligned}
p(k \mid n) &= \int \int \int \frac{p(k, n \mid \lambda,) p(\lambda, \mathcal{R}, x^*)}{p(n)} d\lambda d\mathcal{R} dx^*, \\
&= \int \frac{p(k, n \mid \lambda) p(\lambda)}{p(n)} d\lambda, \\
&= \int p(k \mid n, \lambda) p(\lambda \mid n) d\lambda.
\end{aligned}
\tag{16}
$$

Since $k$ is independent of $n$ given $\lambda$ the probability becomes

$$
p(k \mid n) = \int p(k \mid \lambda) p(\lambda \mid n) d\lambda.
\tag{17}
$$

By applying Bayes theorem, we obtain

$$
p(k \mid n) = \frac{1}{p(n)} \int p(k \mid \lambda) p(n \mid \lambda) p(\lambda) d\lambda.
\tag{18}
$$

Using that $p(\lambda_n)$ is uniform into Equation 18 yield

$$
p(k \mid n) = \frac{1}{p(n)} \int_0^1 p(k \mid \lambda) p(n \mid \lambda) d\lambda.
\tag{19}
$$

Recalling that $p(k \mid \lambda) = \mathcal{B}(1, \lambda)$ and $p(n \mid \lambda) = \mathcal{B}(N, \lambda)$ we have

$$
p(k \mid n) = \frac{\binom{1}{k}\binom{N}{n}}{\int_0^1 p(n \mid \lambda) d\lambda} \int_0^1 \lambda^{k+n}(1-\lambda)^{1-k+N-n} d\lambda.
\tag{20}
$$

Recognizing the Beta distribution we end up with

$$
\begin{aligned}
p(k \mid n) &= \binom{1}{k+1} \frac{\beta(k+n+1, 2-k+N-n)}{\beta(n+1, N-n+1)} \\
&= \frac{\Gamma(k+n+1)\Gamma(2-k+N-n)}{\Gamma(2-k)\Gamma(k+1)\Gamma(n+1)\Gamma(N-n+1)(N+2)}
\end{aligned}
\tag{21}
$$

Finally, we have

$$
p(k \mid n) = \frac{n+1}{N+2} \mathbb{1}(k=1) + \frac{N-n+1}{N+2} \mathbb{1}(k=0).
\tag{22}
$$

$\square$

### A.3. Mira Statistic Law

**Proposition A.3** (Mira statistic law). *Suppose the two random variables $\lambda_n \mid x^*, y_r, c$ and $\lambda_k \mid x^*, y_r, c$ are equal almost surely. Then the random variable $P_N := p(k \mid n)$ converges in distribution to a Beta$(2, 1)$ random variable as $N$ goes to infinity:*

$$
P_N \xrightarrow{d} Beta(2, 1).
\tag{23}
$$

*Proof.* Let $\lambda$ be the common mass of the candidate and true distribution in the random region $\mathcal{R}$, i.e, $\lambda := \lambda_n = \lambda_k$.

By the law of large numbers, we have that the random variable

$$
P_N = \frac{n+1}{N+2} \mathbb{1}(k=1) + \frac{N-n+1}{N+2} \mathbb{1}(k=0)
\tag{24}
$$

converges in probability to

$$P = \lambda \mathbb{1}(k = 1) + (1 - \lambda)\mathbb{1}(k = 0), \tag{25}$$

which implies convergence in distribution. Hence, we have

$$P_N \xrightarrow{d} P. \tag{26}$$

We now need to derive the cumulative distribution function (cdf) of $P$:

$$\begin{aligned} F_P(y) &= \mathbb{P}(P \le y) \\ &= \mathbb{P}(P \le y \mid k = 0)\mathbb{P}(k = 0) + \mathbb{P}(P \le y \mid k = 1)\mathbb{P}(k = 1) \\ &= \mathbb{P}(1 - \lambda \le y \mid k = 0)\mathbb{P}(k = 0) + \mathbb{P}(\lambda \le y \mid k = 1)\mathbb{P}(k = 1) \end{aligned} \tag{27}$$

Recalling that $k$ follows a Bernoulli distribution of parameter $\lambda$ with $\lambda \sim \mathcal{U}[0, 1]$, we derive

$$\mathbb{P}(k = 1) = \int p(k = 1 \mid \lambda)p(\lambda)d\lambda = \int_0^1 \lambda d\lambda = \frac{1}{2}, \tag{28}$$

$$\mathbb{P}(k = 0) = \int p(k = 0 \mid \lambda)p(\lambda)d\lambda = \int_0^1 1 - \lambda d\lambda = \frac{1}{2}. \tag{29}$$

Additionally, using this previous result, $\lambda \sim \mathcal{U}[0, 1]$ and $k \sim \mathcal{B}(\lambda)$, we derive

$$\mathbb{P}(\lambda \le y \mid k = 1) = \int_0^y p(\lambda \mid k = 1)d\lambda = \int_0^y \frac{p(k = 1 \mid \lambda)p(\lambda)}{p(k = 1)}d\lambda = \int_0^y 2\lambda d\lambda = y^2 \tag{30}$$

$$\mathbb{P}(1 - \lambda \le y \mid k = 0) = \int_{1-y}^1 p(\lambda \mid k = 0)d\lambda = \int_{1-y}^1 \frac{p(k = 0 \mid \lambda)p(\lambda)}{p(k = 0)}d\lambda = \int_{1-y}^1 2(1 - \lambda)d\lambda = y^2 \tag{31}$$

The cdf thus is

$$F_P(y) = y^2, \ \forall y \in [0, 1], \tag{32}$$

with the corresponding pdf being $p_P(y) = 2y$, which is exactly the pdf of the Beta(2,1) distribution. $\square$

### A.4. Mira Calibration Score

**Corollary A.4** (Mira statistic moments). *Suppose that $\lambda_n \mid x^*, y_r, c = \lambda_k \mid x^*, y_r, c$ almost surely. Then, the mean and variance of the random variable $P_N$ bounded between 0 and 1, converge to the moments of the Beta(2,1) distribution, that is,*

$$\mathbb{E}_{p(k,n)}[P_N] \to \tfrac{2}{3} \quad \text{as } N \to \infty \tag{33}$$

$$Var_{p(k,n)}[P_N] \to \tfrac{1}{18} \quad \text{as } N \to \infty. \tag{34}$$

*Proof.* The Mira statistic $P_N$ is a random variable bounded between 0 and 1 that converges in distribution to $P \sim Beta(2, 1)$. Since the functions $f(x) = x$ and $g(x) = x^2$ are bounded and continuous on $[0, 1]$, the convergence of the expected values is guaranteed by the definition of weak convergence for bounded functions (a direct consequence of the Portmanteau Theorem, or the Dominated Convergence Theorem). Thus, we have:

$$\mathbb{E}_{p(k,n)}[P_N] \to \mathbb{E}_{p(k,n)}[P] = \frac{2}{3} \quad \text{as } N \to \infty, \tag{35}$$

$$\mathbb{E}_{p(k,n)}\left[P_N^2\right] \to \mathbb{E}_{p(k,n)}\left[P^2\right] = \frac{1}{2} \quad \text{as } N \to \infty. \tag{36}$$

Finally, we derive the variance as

$$\sigma^2_{\text{Mira}}(\mathcal{M}) = \mathbb{E}_{p(k,n)}\left[P_N^2\right] - \mathbb{E}_{p(k,n)}[P_N]^2 \to \mathbb{E}_{p(k,n)}\left[P^2\right] - \mathbb{E}_{p(k,n)}[P]^2 = \frac{1}{18} \quad \text{as } N \to \infty. \tag{37}$$

$\square$

## A.5. Lower bound on the Mira Score

**Proposition A.5** (Lower bound on the Mira score). *For any candidate model $\mathcal{M}$,*

$$\mu_{\text{Mira}}(\mathcal{M}) \geq \frac{1}{2}, \tag{38}$$

*with equality if and only if the candidate and true distributions have disjoint supports.*

*Proof.* Using the law of total expectation, Equation 6 becomes:

$$\mu_{\text{Mira}}(\mathcal{M}) = \mathbb{E}_{p(k,n)}\left[p(k \mid n)\right].$$

Given that $k \mid \lambda_k$ and $n \mid \lambda_n$ are independent we write:

$$p(k,n) = \int p(k \mid \lambda_k) p(n \mid \lambda_n) p(\lambda_n, \lambda_k) d\lambda_n d\lambda_k. \tag{39}$$

Replacing $p(k,n)$ into the expectation, we derive

$$\mathbb{E}_{p(k,n)}\left[p(k \mid n)\right] = \mathbb{E}_{p(\lambda_k, \lambda_n)}\left[\mathbb{E}_{p(k|\lambda_k)p(n|\lambda_n)}\left[p(k \mid n)\right]\right].$$

By substituting the explicit expressions of the Bernoulli and Binomial distributions, we derive

$$\mu_{\text{Mira}}(\mathcal{M}) = \mathbb{E}_{p(\lambda_k, \lambda_n)p(k|\lambda_k)p(n|\lambda_n)}\left[\frac{n+1}{N+2} \cdot \mathbb{1}(k=1) + \frac{N-n+1}{N+2} \cdot \mathbb{1}(k=0)\right]$$

$$= \mathbb{E}_{p(\lambda_k, \lambda_n)}\left[\frac{N\lambda_n+1}{N+2} \cdot \lambda_k + \frac{N-N\lambda_n+1}{N+2} \cdot (1-\lambda_k)\right].$$

After simplifying this expectation, we find:

$$\mu_{\text{Mira}}(\mathcal{M}) = \frac{2N\,\mathbb{E}[\lambda_n\lambda_k] - N\,\mathbb{E}[\lambda_n] - N\,\mathbb{E}[\lambda_k] + N + 1}{N+2}. \tag{40}$$

$$\tag{41}$$

Substituting $\mathbb{E}[\lambda_n] = \frac{1}{2}$ (Proposition 3.1) into Equation 40 and rearranging:

$$\mu_{\text{Mira}}(\mathcal{M}) = \frac{1}{2} + \frac{N}{N+2}\left(2\,\mathbb{E}[\lambda_n\lambda_k] - \mathbb{E}[\lambda_k]\right) \tag{42}$$

Since $N/(N+2) > 0$, the infimum of the Mira score is determined by the infimum of $2\,\mathbb{E}[\lambda_n\lambda_k] - \mathbb{E}[\lambda_k]$.

For fixed $(x^*, c)$, let $F_D$ and $F_{D^*}$ be the radial cumulative distribution functions of the candidate and true distributions around $c$:

$$F_D(r) = \int_{d(y,c)\leq r} p(y \mid x^*, \mathcal{M})\,dy, \qquad F_{D^*}(r) = \int_{d(y,c)\leq r} p(y \mid x^*, \mathcal{M}^*)\,dy. \tag{43}$$

Both are non-decreasing in $r$: as $r$ increases, the ball $\{y : d(y,c) \leq r\}$ grows, and the integral of a non-negative density over it can only increase or stay the same. Since $\mathcal{R}$ has radius $r^* = d(y_r, c)$, by definition:

$$\lambda_n = F_D(r^*), \qquad \lambda_k = F_{D^*}(r^*). \tag{44}$$

We let $F_D^{-1}(\lambda_n) = \inf\{r \geq 0 : F_D(r) \geq \lambda_n\}$ be the generalized inverse, which is well-defined and non-decreasing for any non-decreasing $F_D$, and define the map

$$T_{x^*,c}(\lambda_n) = F_{D^*}\big(F_D^{-1}(\lambda_n)\big) \tag{45}$$

which is non-decreasing as a composition of two non-decreasing functions. Moreover, $F_D^{-1}(\lambda_n) = r^*$ almost surely: $F_D$ is strictly increasing at every realized $r^*$ (since $r^*$ falls in the support of $D = d(y,c)$ with $y \sim p(y \mid x^*, \mathcal{M})$, where the

density is positive), and $F_D$ may only be flat on intervals where the candidate density is zero, which $r^*$ visits with probability zero. Therefore $\lambda_k = T_{x^*,c}(\lambda_n)$ almost surely.

By Proposition 3.1, $\lambda_n \sim \mathcal{U}[0, 1]$, so the conditional expectations for fixed $(x^*, c)$ are:

$$\mathbb{E}[\lambda_n \lambda_k \mid x^*, c] = \int_0^1 \lambda_n \cdot T_{x^*,c}(\lambda_n) \, d\lambda_n, \qquad \mathbb{E}[\lambda_k \mid x^*, c] = \int_0^1 T_{x^*,c}(\lambda_n) \, d\lambda_n. \tag{46}$$

Since $\lambda_n$ and $T_{x^*,c}(\lambda_n)$ are both non-decreasing on $[0, 1]$, Chebyshev's integral inequality gives:

$$\int_0^1 \lambda_n \cdot T_{x^*,c}(\lambda_n) \, d\lambda_n \geq \int_0^1 \lambda_n \, d\lambda_n \cdot \int_0^1 T_{x^*,c}(\lambda_n) \, d\lambda_n = \frac{1}{2} \int_0^1 T_{x^*,c}(\lambda_n) \, d\lambda_n, \tag{47}$$

and therefore:

$$2 \, \mathbb{E}[\lambda_n \lambda_k \mid x^*, c] - \mathbb{E}[\lambda_k \mid x^*, c] = \int_0^1 (2\lambda_n - 1) \, T_{x^*,c}(\lambda_n) \, d\lambda_n \geq 0. \tag{48}$$

Since this holds for every $(x^*, c)$, marginalizing over $(x^*, c) \sim p(x \mid \mathcal{M}^*) \, p(c)$ and substituting into (42):

$$\mu_{\mathrm{Mira}}(\mathcal{M}) = \frac{1}{2} + \frac{N}{N + 2} \underbrace{\left( 2 \, \mathbb{E}[\lambda_n \lambda_k] - \mathbb{E}[\lambda_k] \right)}_{\geq 0} \geq \frac{1}{2}. \tag{49}$$

Equality requires $\int_0^1 \lambda_n \cdot T_{x^*,c}(\lambda_n) \, d\lambda_n = \frac{1}{2} \int_0^1 T_{x^*,c}(\lambda_n) \, d\lambda_n$, for almost all $(x^*, c)$. This equality holds if and only if $T_{x^*,c}(\lambda_n)$ is constant (see Theorem 1.2 of Jakubowski, 2021). Because $T_{x^*,c}(0) = 0$, $T_{x^*,c}$ is 0 almost everywhere on $[0, 1]$, meaning the true distribution assigns zero mass to every ball defined by the candidate. This corresponds to the case where the candidate and true conditional distributions have disjoint supports.

$\square$

## B. Rank-Ordering Reformulation of Mira and Sufficiency Condition

We have shown in A.3 that the random variable $P_N := p(k \mid n)$ being $Beta(2, 1)$ distributed and the Mira score converging to $2/3$ in the large $N$ limit are necessary conditions for the proposal and the reference distributions to be equal. While neither is a sufficient condition, we can define alternative Mira statistics and score that only consider the cases where the reference sample falls inside the ball, and show that the corresponding random variable $q_N := p(k \mid n)$ when $k = 1$ for this modified statistics being $B(2, 1)$ distributed is a sufficient condition for calibration.

We present this sufficiency proof here as opposed to the main text since, because the new definition involves only looking at the cases where $y^*$ falls inside the ball, it becomes far less sample efficient. Since we expect the main use case for Mira to be in sample-limited regimes, we expect the method as presented in the body of the paper to be most commonly used. However, for completeness and because it might be useful in cases where guarantees of accuracy are important, we present the proof of sufficiency for the alternative Mira definition here.

We start in Section B.1 by deriving the main results of the paper in term of rank, then, in Section B.2 we present the sufficiency proof.

### B.1. Alternative Framing of Mira in Terms of Rank-Ordering

In this subsection, we start by reframing Mira in the language of ranks of samples. For any given conditional we want to test, for every value of $P_N$ we calculate, we have a set of $N + 2$ samples: one samples from the the reference true distribution (the ground truth $y^*$) and $N + 1$ i.i.d. samples from the proposal conditional distribution to test. Since we remove $y_r$ from the $N + 1$ samples from the proposal conditional distribution, we are left with $N$ samples to classify as 'in' or 'out' of the ball.

We start by ranking all $N + 1$ samples from the proposal conditional distribution based on their distance to the center $c$.

The number of samples inside the ball, $n$, is one minus the rank of $y_r$ among the remaining $N$ samples, since it is the number of samples that are closer to $c$ than $y_r$ (excluding $y_r$ itself). Since the $N + 1$ samples are i.i.d. samples from the

same distribution, $y_r$ is equally likely to have any rank. This means that $n$ is a random integer uniformly distributed on $\{0, 1, 2, ..., N\}$.

The sample $y^*$ is an $(N + 2)$-th sample whose position in the rank we want to compare to the region defined by $y_r$.

**Calculation of the Mira score theoretical value for finite $N$ under the null, and asymptotic value (Corollary 3.4)**

Under the null hypothesis that the proposal samples and the reference sample are drawn from the same distribution, we can easily get the probability of $y^*$ being in the ball, $p(k = 1 \mid n)$, and the probability of it being outside the ball, $p(k = 0 \mid n)$, by a simple counting argument: The $N + 1$ ranked samples by their distance to $c$ form $N + 2$ 'positions' where we can insert $y^*$ in the rank. Under the null hypothesis that $y^*$ is drawn from the same distribution, it is equally likely to fall into any of these $N + 2$ positions. The ball's boundary is at $y_r$, which has rank $n + 1$. Therefore, the probability that $y^*$ lands in one of the $n + 1$ slots between $c$ and $y_r$ ($k = 0$) is

$$p(k = 1 \mid n) = (n + 1)/(N + 2) \,. \tag{50}$$

Meanwhile, the probability that $y^*$ is outside the ball ($k = 0$) is:

$$p(k = 0 \mid n) = 1 - p(k = 1 \mid n) = (N - n + 1)/(N + 2) \,. \tag{51}$$

We can write the Mira score as the expectation of $P_N := p(k \mid n)$:

$$\mu_{\mathrm{Mira}}(\mathcal{M}) = \mathbb{E}_{p(n)} \left[ \mathbb{E}_{p(k|n)} \left[ P_N \right] \right] \tag{52}$$

$$= \frac{1}{N + 1} \sum_{n=0}^{N} \left( p_{in}(n + 1)/(N + 2) + p_{out}(N - n + 1)/(N + 2) \right) \tag{53}$$

$$= \frac{2N + 3}{3(N + 2)} \,, \tag{54}$$

where we used $p(n) = 1/(N + 1)$. Therefore, the expected value of $P_N$ in the asymptotic limit is

$$\lim_{N \to \infty} \mu_{\mathrm{Mira}}(\mathcal{M}) = 2/3 \,. \tag{55}$$

**Alternative demonstration of Mira statistics law (Proposition 3.3)**

More generally, we can also show that the values of $P_N$ are $Beta(2, 1)$ distributed under the null hypothesis using the rank formulation. To do so, we define $J$ and $K$ as the normalized ranks (by $N + 2$) of $y^*$ and $y_r$, respectively, which have support on $[0, 1]$ when $N \to \infty$. Because all the samples are assumed to be i.i.d., $J$ and $K$ are random variables uniformly distributed on this interval.

With this notation, we can write the values of $P_N$ for a given trial:

$$P_N = \begin{cases} p(k = 1 \mid n) = K & \text{if } J < K \,, \\ p(k = 0 \mid n) = 1 - K & \text{if } J > K \,. \end{cases} \tag{56}$$

We wish to find the pdf of the variable $P_N$. We do this by calculating its cumulative distribution function (CDF), $F(x) = p(P_N \le x)$, using a geometric argument.

In the large $N$ limit, the possible values of $(J, K)$ define a unit square. Since $J$ and $K$ are uniform, the probability of $(J, K)$ falling into any specific area is simply the area of that region.

- If $y^*$ is in the region, $J < K$:
  We want the probability that both $P_N \le x$ and $J < K$. Since, in this case, $P_N = K$, this is the area of the region in the unit square where $0 \le J < K$ and $K \le x$. This is a triangle with area $(1/2) \cdot x \cdot x = x^2/2$.

- If $y^*$ is outside the region, $J > K$:
  We want the probability that both $P_N \le x$ and $J > K$. Since, in this case, $P_N = 1 - K$, this is the area of the region in the unit square where $K < J \le 1$ and $K \ge 1 - x$. This is, again, simply a triangle with area $(1/2) \cdot (1 - (1 - x)) \cdot (1 - (1 - x)) = x^2/2$.

The total probability that $P_N$ is less than this value of $x$, $p(P_N \leq x)$, is the sum of these two possible cases. Therefore, we have that the CDF of $p(P_N)$ is

$$F(x) = x^2 \,. \tag{57}$$

The PDF, $p(P_N)$, can then easily be found by taking the derivative of $F(x)$:

$$p(x) = \frac{d(x^2)}{dx} = 2x \qquad \text{for} \quad x \in [0,1] \,. \tag{58}$$

Moreover, $p(P_N) = 2P_N$, is precisely the PDF of a $Beta(2,1)$ distribution. This distribution has an average of $2/3$, $\mathbb{E}(P_N) = 2/3$, and the variance of $P_N$ is $Var(P_N) = 1/18$.

## B.2. Alternative Definition of Mira and Sufficiency Proof

As mentioned above, we can define alternative Mira statistics and score that only considers the cases where the reference sample falls inside the ball, and show that the corresponding random variable $q_N := p(k \mid n)$ when $k = 1$ being $Beta(2,1)$ distributed is a necessary and sufficient condition for calibration.

**$q_N$ being $Beta(2,1)$ distributed is a necessary condition of calibration.**

First, we prove the necessary direction, that is, we'd like to show that $q_N$ is distributed according to a $Beta(2,1)$ when we only consider the $k = 1$ cases, under the assumption that the underlying distributions are the same.

With this alternative definition, we only consider the cases where $q_N = p_{in} = (n+1)/(N+2)$ (using the same rank argument as above). To average this over possible values of $n$ knowing that $k = 1$, we must find $p(n \mid k = 1)$ to calculate:

$$\mathbb{E}_{p(k=1|n)}\left[q_N\right] \,. \tag{59}$$

Using Bayes' theorem,

$$p(n \mid k = 1) = \frac{p(k = 1 \mid n)p(n)}{p(k = 1)} \,, \tag{60}$$

$$= \frac{2(n+1)}{(N+1)(N+2)} \,. \tag{61}$$

We have used that $p(n) = \frac{1}{N+1}$ and that

$$p(k = 1) = \mathbb{E}_{p(n)}\left[p(k = 1 \mid n)\right] = \frac{1}{1+N}\sum_{n=0}^{N} p(k = 1 \mid n) = \frac{1}{2} \,. \tag{62}$$

We therefore find that the value of the Mira score tends to

$$\lim_{N \to \infty} \mu_{\text{Mira}}(\mathcal{M}) = \lim_{N \to \infty}\left[\mathbb{E}_{p(n|k=1)}\left[q_N\right]\right] \tag{63}$$

$$= \lim_{N \to \infty}\left[\frac{2}{3}\frac{(N+3/2)}{(N+2)}\right] = 2/3 \,, \tag{64}$$

as before.

To show that $q_N$ is $Beta(2,\ 1)$ distributed in the asymptotic limit, we define the normalized ranks $J$ and $K$ of $y^*$ and $y_r$ as above. The values of $q_N$ are given by:

$$q_N := p(k = 1 \mid n) = K \qquad \text{since } J < K \,. \tag{65}$$

To find the PDF of $q_N$, we calculate the CDF $F(x) = p(q_N \leq x)$ using a geometric argument as above, integrating the area in $(J,\ K)$ space where $y^*$ is inside the ball. This is the area of the triangle defined by $0 < K < J$ and $K < x$, which has area $x^2/2$. Normalizing to make this a CDF on [0,1], this means the CDF of $p(q_N)$ is $F(x) = x^2$, and

$$p(q_N) = \frac{d(q_N^2)}{dq_N} = 2q_N \qquad \text{for} \quad q_N \in [0,1] \,, \tag{66}$$

which is again the PDF of a $Beta(2, 1)$ distribution.

**$q_N$ being $Beta(2, 1)$ distributed is a sufficient condition for calibration.**

Conversely, to prove the sufficiency condition, we assume the probability $q_N$ we obtain through the alternative Mira algorithm are asymptotically distributed according to a $Beta(2, 1)$ distribution. We define $K$ and $J$ as above, and let $f(K, J)$ be their joint distribution on $[0, 1]$. To show that the reference and the proposal distribution are identical, it is enough to show that $f(K, J)$ is uniform on the unit square, that is, $K$ and $J$ must be uniformly distributed on $[0, 1]$. As before, we know that

$$q_N = p(k = 1 \mid n) = K \qquad \text{since } J < K. \tag{67}$$

Since the $q_N$ are distributed as a $Beta(2, 1)$ distribution by assumption, their CDF must be $F(x) = x^2$. As before, this can also be calculated from $f(K, J)$ by integrating the probability that $J < K$ and $q = K \leq x$ for a given value of $x$:

$$F(x) = \int_0^x dK \int_0^K dJ \, [f(K, J)] = x^2. \tag{68}$$

Differentiating with respect to $x$ we obtain, using the fundamental theorem of calculus:

$$\frac{d}{dx} F(x) = 2x = \int_0^x dJ \, [f(x, J)]. \tag{69}$$

Now, we know the normalized rank $K$ must be uniform, since it is the rank of $y_r$ among $i.i.d.$ samples from the same distribution. Therefore, $f(K, J) = f_K(K) f_J(J) = C \, f_J(J)$, with $C$ a constant, and we find:

$$2x = C \int_0^x dJ \, [f_J(J)], \tag{70}$$

$$= C \, (F_J(x) - F_J(0)), \tag{71}$$

$$= C \, F_J(x), \tag{72}$$

where $F_J$ denotes the CDF of $f_J(J)$. Since this must hold for all $x$, we have that $f_J(x)$ must be uniform on $[0, 1]$ for any value of $K$, and therefore the joint $f(J, K)$ must be uniform on the unit square, which completes the proof.

## C. Mira Interpretability

In this section, we show that the Mira score can provide interpretability in terms of over- and underconfidence.

Starting from Equation 40, with $\mathbb{E}[\lambda_n] = 1/2$ (Proposition 3.1) and recognizing that $2\,\mathbb{E}[\lambda_n \lambda_k] - \mathbb{E}[\lambda_k] = 2\mathrm{Cov}(\lambda_n, \lambda_k)$:

$$\mu_{\mathrm{Mira}}(\mathcal{M}) \xrightarrow{N \to \infty} \frac{1}{2} + 2\,\mathrm{Cov}(\lambda_n, \lambda_k).$$

Under the null hypothesis ($\lambda_k = \lambda_n$ a.s.), this gives $\mu = 1/2 + 2\,\mathrm{Var}(\lambda_n) = 2/3$ (Corollary 3.4). Deviations from $2/3$ are thus governed by how $\mathrm{Cov}(\lambda_n, \lambda_k)$ compares to $\mathrm{Var}(\lambda_n)$.

Since $\mathrm{Var}(\lambda_n) = 1/12$ is fixed (Proposition 3.1), the key quantity is $\mathrm{Var}(\lambda_k)$: how much the true mass varies across random regions. An overconfident candidate concentrates its regions in a limited area of the true distribution, so $\lambda_k$ sees only a restricted range of true mass values, yielding $\mathrm{Var}(\lambda_k) < \mathrm{Var}(\lambda_n)$. Conversely, an underconfident candidate probes the full extent of the truth, and $\lambda_k$ swings from 0 to 1, yielding $\mathrm{Var}(\lambda_k) > \mathrm{Var}(\lambda_n)$.

When $\mathrm{Var}(\lambda_k) < \mathrm{Var}(\lambda_n)$, By Cauchy Schwarz we derive $\mathrm{Cov}(\lambda_n, \lambda_k) < \mathrm{Var}(\lambda_n)$, hence $\mu < 2/3$. This motivates defining:

- **Overconfident**: $\mathrm{Var}(\lambda_k) < \mathrm{Var}(\lambda_n) \Longrightarrow \mu_{\mathrm{Mira}} < 2/3$ (by Cauchy Schwarz).

- **Underconfident**: $\mu_{\mathrm{Mira}} > 2/3 \Longrightarrow \mathrm{Var}(\lambda_k) > \mathrm{Var}(\lambda_n)$ (contrapositive).

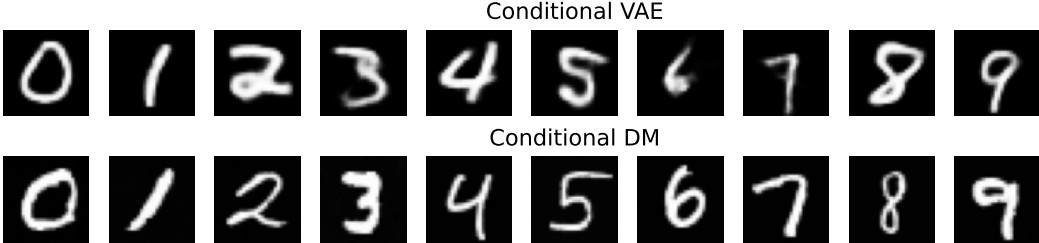

*Figure 6.* **Top row:** Samples generated by the conditional VAE for a single draw of digits $\{1, \ldots, 9\}$. In this particular realization, digits 3, 6, and 7 exhibit noticeable distortions, while the remaining digits are reasonably well formed, suggesting mild limitations in the model's ability to capture the target conditional distribution. **Bottom row:** Samples generated by the conditional diffusion model for the same conditioning, where all digits appear sharp and correctly rendered, indicating a stronger capacity to model the conditional distribution. Although based on a single qualitative example, these observations are consistent with the trends captured by the Mira score reported in Figure 3.

This formulation clarifies the interpretation of score differences in experiment of section 5.1: a score of $\sim 0.62$ is consistent with overconfidence, while a score of $\sim 0.7$ necessarily requires $\mathrm{Var}(\lambda_k) > \mathrm{Var}(\lambda_n)$, consistent with underconfidence. These two scores thus reflect qualitatively different failure modes despite similar distances from $2/3$. The Mira score thus provides not only quantitative fidelity assessment but also qualitative diagnostic information about the nature of misspecification.

# D. Experiments Details

## D.1. Conditional Distribution Experiment

In this section, we provide additional details for the experiments in Section 5.2.

### D.1.1. ARCHITECTURE AND HYPERPARAMETERS

The diffusion model was implemented using the `score-models`[2] package. It utilizes an NCSN++ architecture (Song et al., 2021) with a variance exploding noising process. The model was trained with the Adam optimizer (Kingma & Ba, 2015) ($lr = 1 \times 10^{-4}$, batch size 256, `ema_decay = 0.999`). All other hyperparameters followed the `score-models` defaults. Training was performed on an A100 GPU for $\sim$2 hours (wall time) using 32 GB of VRAM.

The conditional VAE used a convolutional encoder with layer structure $(28 \times 28) \rightarrow (32 \times 14 \times 14) \rightarrow (64 \times 7 \times 7) \rightarrow 32$, and a symmetric decoder with transposed convolutions. Class conditioning was introduced through learned label embeddings concatenated to both the encoder and decoder inputs. The model was trained for 500 epochs with batch size 512 using Adam ($lr = 3 \times 10^{-4}$, weight decay $10^{-5}$) and a cosine annealing learning rate schedule, with KL warmup.

### D.1.2. GENERATED SAMPLES

In Figure 6, we show generated samples (a single draw of digits $\{0, \ldots, 9\}$) for the two conditional generative models. For both models, we directly sample from the learned conditional distribution $p(y \mid x)$ given the class label $x$ where each sample is drawn independently according to the respective model's conditional generative mechanism. Although only one sample per class is displayed, the conditional diffusion model produces sharper and more accurate digits than the conditional VAE. This qualitative observation is consistent with the trends captured by the Mira score reported in Figure 3.

## D.2. Physical Model Mismatch Experiment

We provide details about the physical forward models used in Section 5.3.1 and the sampling to get posteriors.

### D.2.1. FORWARD MODELS

Following Filipp et al. (2025), we generate strong lensing images with background sources composed of one or three Sérsic components (Sérsic, 1963), allowing us to vary the complexity of source morphologies in a controlled manner. A Sérsic

---

[2]github.com/AlexandreAdam/score_models

*Table 1.* Parameter ranges for lens and source parameters that are inferred. All other parameters are held constant across models. SIE models implicitly fix $\gamma = 2.0$.

| Parameter | Distribution |
|---|---|
| **EPL Lens** | |
| Einstein radius $b$ | $\mathcal{U}[1.0, 1.5]$ |
| Axis ratio $q$ | $\mathcal{U}[0.5, 0.9]$ |
| Orientation angle $\phi$ | $\mathcal{U}[0.0, \pi]$ |
| Power-law slope $\gamma$ | $\mathcal{U}[1.75, 2.25]$ |
| **SIE Lens** | |
| Einstein radius $b$ | $\mathcal{U}[1.0, 1.5]$ |
| Axis ratio $q$ | $\mathcal{U}[0.5, 0.9]$ |
| Orientation angle $\phi$ | $\mathcal{U}[0.0, \pi]$ |
| **Sérsic Source** | |
| Source center $\hat{x}_{\mathrm{src}}$ | $\mathcal{U}[-0.5, 0.5]$ |
| Source center $\hat{y}_{\mathrm{src}}$ | $\mathcal{U}[0.05, 0.10]$ |
| Effective intensity $I_e$ | $\mathcal{U}[0.4, 0.8]$ |

profile is a parametric model for a galaxy's surface brightness, with parameters controlling its size, ellipticity, orientation, and concentration. For the lens mass distribution, we consider either an EPL or an SIE model. All lensing images are generated using `caustics` (Stone et al., 2024). The resulting lensed galaxy images are defined on a $100 \times 100$ grid with a pixel size of $0.05$ arcsec. Independent Gaussian noise with distribution $\mathcal{N}(0, 1)$ is then added to each pixel. Table 1 summarizes the parameters of the forward models that we aim to infer. Figure 7 provides an example of the observed image (second column) and image without noise (first column) of the true forward model (EPL + 3 Sérsic sources).

### D.2.2. POSTERIOR SAMPLES

To get the posterior samples, MALA is run using 100 walkers, with 200 steps for burn-in and 200 steps for sampling, yielding $20\,000$ posterior samples per model. In setups with three Sérsic sources, we independently sample source-specific parameters, i.e., the position $(\hat{x}_{\mathrm{src}}, \hat{y}_{\mathrm{src}})$, and the intensity $(I_e)$, for each component. MALA was run on a single AMD Milan CPU core for approximately 8 minutes (wall-time) per configuration, using up to 10 GB of memory. Across 100 synthetic observations, the total inference cost was approximately 13.33 CPU-hours.

When running Mira, all models are evaluated in the 13-dimensional parameter space. For single-source configurations, the second and third source positions $(\hat{x}_{\mathrm{src}}, \hat{y}_{\mathrm{src}})$ and intensities $(I_e)$ are fixed to zero, ensuring a consistent parameter structure across models and allowing for fair posterior comparisons.

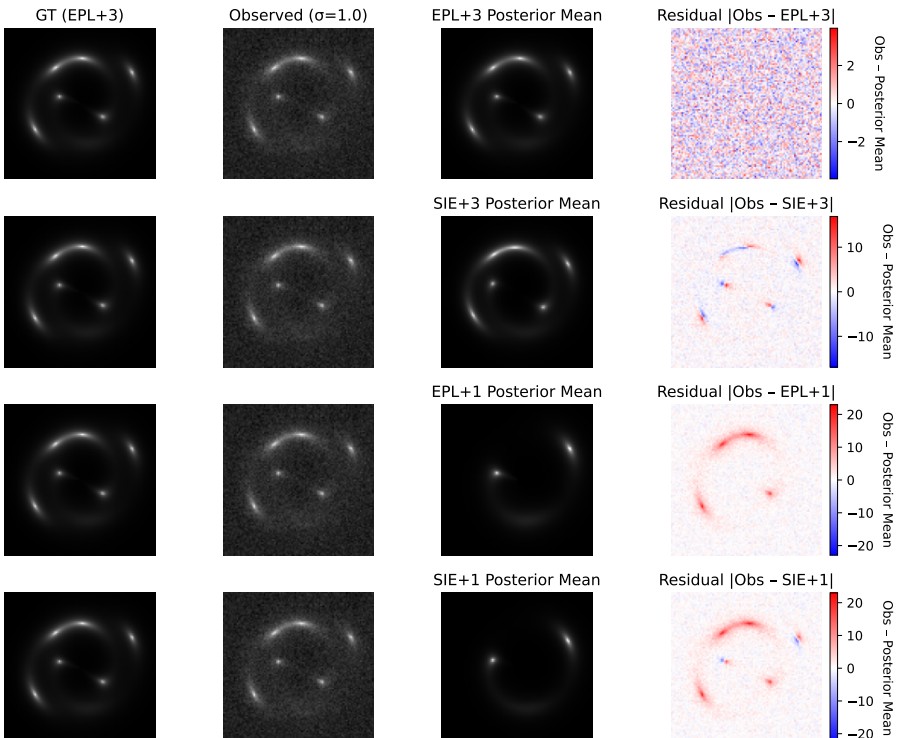

*Figure 7.* Left to right: clean ground truth image (EPL + 3 Sérsic sources), observed data with Gaussian noise ($\sigma = 1$), posterior means from four candidate models, and corresponding residuals (observation minus posterior mean). Only the correctly specified model (top row: EPL + 3 Sérsic source) produces residuals consistent with Gaussian noise. Other models show structured residuals, revealing mismatches due to incorrect lens type and/or source count.

Figure 7 shows one example of the clean ground truth image (EPL + 3 Sérsic sources), the corresponding noisy observation, and posterior means from each candidate model: EPL+3, SIE+3, EPL+1, and SIE+1. Each posterior mean is computed by averaging 100 MALA samples. The final column shows residuals between the observation and the posterior mean. Only the correctly specified model (EPL + 3 Sérsic, top row) produces residuals consistent with Gaussian noise. All other models exhibit structured residuals, revealing mismatches due to incorrect lens profile, source count, or both. Among these models, the SIE + 3 Sérsic model yields smaller residual amplitudes than the single source models. The EPL + 1 Sérsic and SIE + 1 Sérsic models exhibit residuals of comparable magnitude. This ordering is consistent with the ranking provided by the Mira score.

### D.3. Prior Distribution and Noise Model Shifts Experiment

In this section, we provide details about the physical forward models used in Section 5.3.2 and the sampling to get posteriors.

#### D.3.1. ARCHITECTURE AND HYPERPARAMETERS

In this experiment, we employ a score-based diffusion model to learn the prior distribution from images of elliptical and spiral galaxies. We then combined this learned prior with our analytical likelihood that lens the input source galaxy to get posterior samples (see Barco et al., 2025 for further details). The models were implemented using the `score-models` package and adopt an NCSN++ architecture (Song et al., 2021). Training was performed using the Adam optimizer (Kingma & Ba, 2015), with a learning rate of $1 \times 10^{-4}$, a batch size of 256, and an exponential moving average decay parameter `ema_decay = 0.999`, for approximately $2.5 \times 10^5$ optimization steps. All hyperparameters not explicitly specified were set to the default values provided by the `score-models` package. Each model was trained on a single NVIDIA A100 GPU for roughly 20 hours of wall-clock time, with 32 GB of allocated VRAM.

$$y = A\mathbf{x}^\star + \eta \qquad \mathbf{x}^\star \qquad \mathbf{x} \sim p_0(\mathbf{x} \mid \mathbf{y}) \ \ \mathbf{x} \sim p_1(\mathbf{x} \mid \mathbf{y}) \ \ \mathbf{x} \sim p_2(\mathbf{x} \mid \mathbf{y}) \ \ \mathbf{x} \sim p_3(\mathbf{x} \mid \mathbf{y})$$

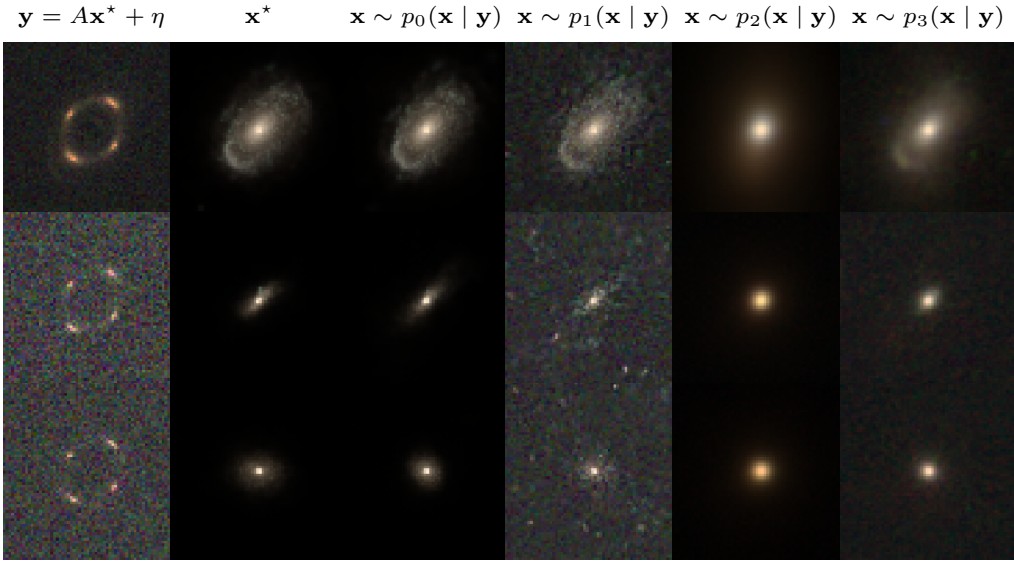

*Figure 8.* Plotted in order of left to right is the result of the forward model noised up. The 2nd column is the ground truth, next is the first posterior model with elliptical galaxy prior and $\sigma_n = 2.0$, the next column is the posterior given elliptical galaxy prior and $\sigma_n = 2$, the 5th column is the posterior model with a spiral galaxy prior and $\sigma_n = 0.5$, and lately the last column is the posterior model given a spiral galaxy prior and $\sigma_n = 0.5$. Rows are different sources.

#### D.3.2. POSTERIOR SAMPLES

To get posterior samples, we use the same SDE solver setup for prior and posterior sampling as (Barco et al., 2025), which is a predictor-corrector solver (Song et al., 2021) with 1024 solver steps. We obtain 16 prior samples to simulate the ground truths $x^*$, and get 64 posterior samples per observation per configuration. Inference of these 4 112 samples was carried out in a single A100 GPU for 4 hours (wall-time) and 40Gb of VRAM allocated.

In Figure 8, we show the true source galaxy (second column) and the corresponding lensed and noisy observation (first column). We then display one posterior sample for each candidate model. Consistent with the Mira score, the best

reconstruction of the source galaxy is obtained when using the correct prior and noise model (third column), followed by the model with the correct prior but an incorrect noise model (fourth column). The two remaining models, which rely on an incorrect prior, both yield poor reconstructions, and their relative performance is difficult to distinguish qualitatively.

## E. Additional Experiments

In this section we provide additional experiments on which we test Mira.

### E.1. InverseBench

We build on the recently introduced INVERSEBENCH framework (Zheng et al., 2025), which evaluates plug-and-play diffusion priors (PnPDPs) across a diverse set of scientific inverse problems, including compressed sensing MRI, black hole imaging, and linear inverse scattering. For full experimental details, we refer the reader to Zheng et al., 2025. While InverseBench highlights the promise of diffusion models in producing high-fidelity reconstructions, its evaluation depends on deterministic metrics such as PSNR, SSIM, and task-specific misfit scores such as $\tilde{\chi}^2$. These metrics assess image quality and data consistency, but do not evaluate the calibration of the inferred posterior. For this reason, we are not using these metrics to validate Mira.

Across all experiments, we compare the following models: DPS (Chung et al., 2023), REDDiff (Mardani et al., 2024), DiffPIR (Zhu et al., 2023a), DAPS (Zhang et al., 2025a), and PnP-DM (Wu et al., 2024). We consider $L = 8$ true parameters, and for each, we generate $N = 50$ i.i.d. posterior samples.

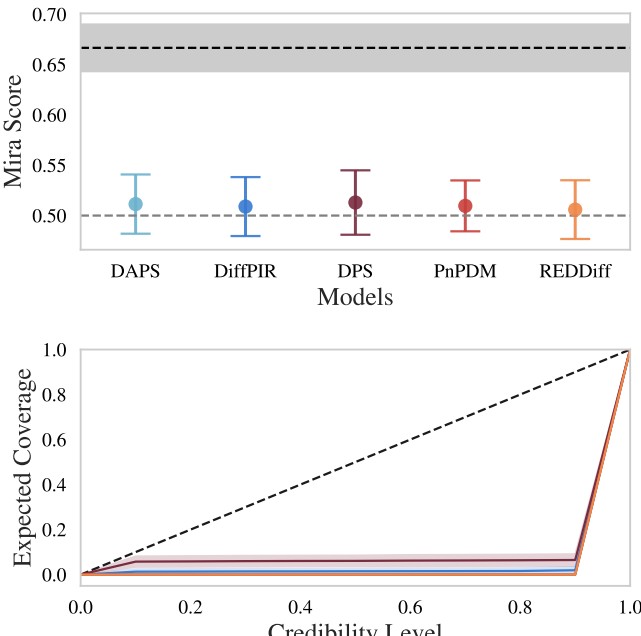

*Figure 9.* **Top**: Mira applied to the Black Hole Imaging inverse problem shows all models are uncalibrated, with DPS performing best. The shaded gray band indicates the theoretical uncertainty, $2/3 \pm \sqrt{1/18L}$. **Bottom**: TARP results are consistent with Mira 's assessment.

### E.1.1. BLACK HOLE IMAGING

The black hole imaging experiment aims to recover ideal $64 \times 64$ pixels images, $\mathbf{z}$, of black hole event horizon in position space from measured $\mathbf{I}_{a,b}^t(\mathbf{z})$ representing the Fourier component of the image. Visibilities are then corrupted by the instrument measurement as follows:

$$V_{a,b}^t = g_a^t g_b^t e^{-i(\phi_a^t - \phi_b^t)} \mathbf{I}_{a,b}^t(\mathbf{z}) + \eta_{a,b}^t, \tag{73}$$

with (a,b) denoting a pair of telescopes, $\eta_{a,b}^t$ Gaussian thermal noise, $g_a^t$ and $g_b^t$ the telescope-dependent amplitude errors, and $\phi_a^t$ the phase errors $\phi_a^t, \phi_b^t$.

Using this forward model, we perform posterior inference with pretrained diffusion priors, generating multiple samples

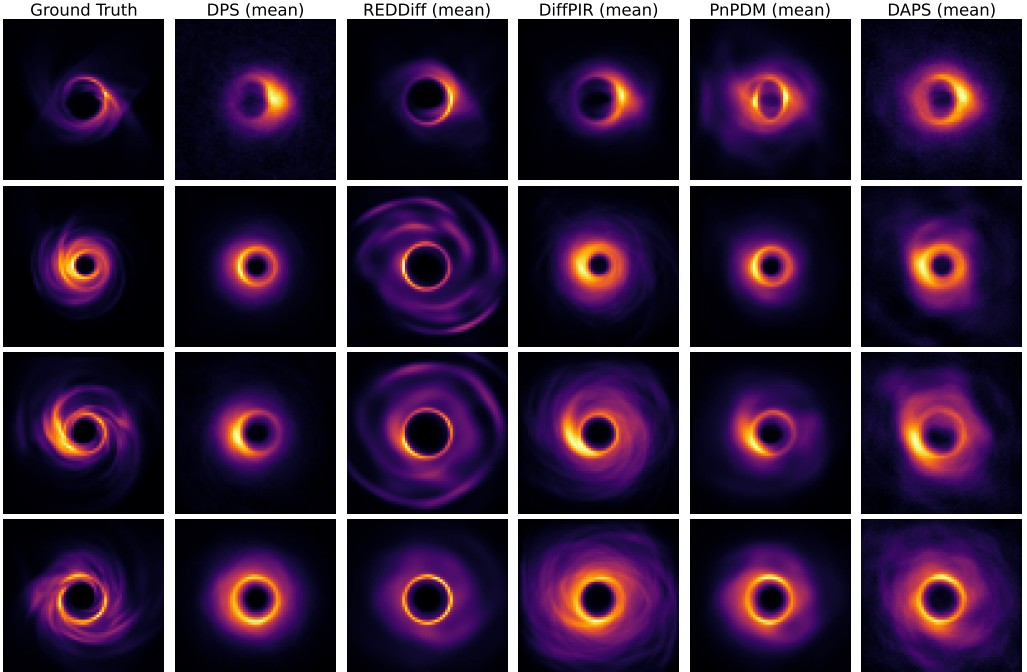

*Figure 10.* The mean of the posterior samples from the black hole imaging inverse problem. All models struggle to correctly recover the truth black hole image.

per test image. We then evaluate these posterior samples using Mira to quantify how well they capture the true posterior distribution. Figure 9 shows Mira scores for the black hole imaging inverse problem. Across all methods, scores remain close to the lower bound of $1/2$, indicating that none of the evaluated models reliably capture the posterior structure. This aligns with prior findings that black hole imaging is an ill-posed problem (Zhang et al., 2025b; Leong et al., 2023; Wu et al., 2024). TARP confirms the Mira findings. Figure 10 shows the means of the 50 black hole image samples we generate.

### E.1.2. COMPRESSED SENSING MRI

The compressed sensing MRI experiment aims to recover the image $\mathbf{z} \in \mathbb{C}^m$ of dimension $320 \times 320$ pixels, from the following corrupted observation

$$\mathbf{y}_j = \mathbf{PFS}_j\mathbf{z} + \mathbf{n}_j, \quad \text{for } j = 1, \ldots, J, \tag{74}$$

where $\mathbf{P}$ is the subsampling mask, $\mathbf{F}$ is the Fourier transform, $\mathbf{S}_j$ is the sensitivity map of the j-th coil, and $\mathbf{n}_j$ is measurement noise.

Figure 11 presents Mira scores, and Figure 12 displays the mean of the 50 posterior samples per model. While most models achieve strong reconstruction fidelity, their posterior samples are poorly calibrated. Notably, DPS achieves a Mira score of 0.58, indicating significantly better calibration compared to the other models. Again, TARP confirms the Mira findings.

These results highlight that fidelity metrics alone cannot identify when models fail to capture uncertainty correctly. On the other hand, coverage tests such as TARP capture the uncertainty but do not enable ranking posteriors. Mira offers a complementary evaluation, enabling principled comparisons of posterior quality across generative inference pipelines.

### E.2. Bayes Factor Comparison

In this appendix, we compare Mira with the Bayes factor (BF) on two experiments. We do not include this comparison in the main experiments because computing the BF is not tractable in that setting. Instead, we construct toy problems for which the evidence can be computed exactly.

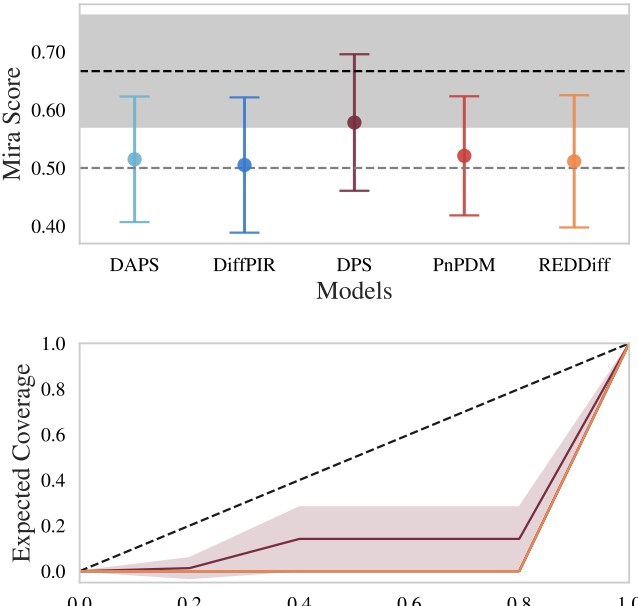

*Figure 11.* **Top**: Mira applied to compressed sensing MRI identifies DPS as the best-calibrated model, though all models remain miscalibrated. Horizontal lines denote expected behavior: well-calibrated (black), underconfident (upper gray), and overconfident or biased (lower gray). The shaded gray band indicates the theoretical uncertainty, $2/3 \pm \sqrt{1/18L}$. Note that the theoretical variance is large due to there only being 6 fiducial samples to perform inference on. **Bottom**: TARP supports this conclusion, DPS performs best, but no model achieves calibration.

### E.2.1. LINEAR REGRESSION

We start by considering a toy Bayesian inference problem where we aim to infer the posterior distribution $p(y \mid x)$ over weights $y = [m, b] \sim \mathcal{N}([0, 0], [0.5, 2])$ of the linear model $x = mz + b + \eta$ with $\eta \sim \mathcal{N}(0, \sigma^2)$ and $z \sim \mathcal{U}[-1, 1]$. We define five models where we increase the noise levels, $\sigma = \{0.001, 0.01, 0.01, 0.10, 0.20, 0.25\}$, and choose the model with the least noise as our true model.

To compute Mira, for each model, we consider $L = 5\,000$ true samples and consider 100 regions per true sample. For each of these observations, we draw $N = 5\,000$ samples from our analytic posterior distribution. The BF is defined as the ratio of the marginal likelihood of a candidate model to that of a reference (true) model:

$$\text{BF}(\eta) = \frac{p(y \mid \mathcal{M}_\eta)}{p(y \mid \mathcal{M}^*)},$$

where $\mathcal{M}_\eta$ denotes the model parameterized by the noise level $\eta$, and $\mathcal{M}^*$ denotes the model associated with the smallest noise level ($\eta = 0.001$), which is taken as the ground-truth data-generating model. To make the results consistent with Mira, we average the BF over the $L = 5\,000$ true observations.

*Table 2.* Comparison of Mira and BF scores in linear regression. Mira and BF both rank models consistently with increasing levels of misspecification. Mira scores range from $1/2$ (poorly specified model) to $2/3$ (well specified model). BF near 1 indicate models that are nearly as plausible as the reference model $\mathcal{M}^*$; lower values indicate less support.

| Noise Level | Mira | BF |
|---|---|---|
| **0.001** | **0.6638** | **0.999** |
| 0.01 | 0.6422 | 0.990 |
| 0.1 | 0.5668 | 0.906 |
| 0.15 | 0.5589 | 0.863 |
| 0.2 | 0.5551 | 0.821 |
| 0.25 | 0.5521 | 0.782 |

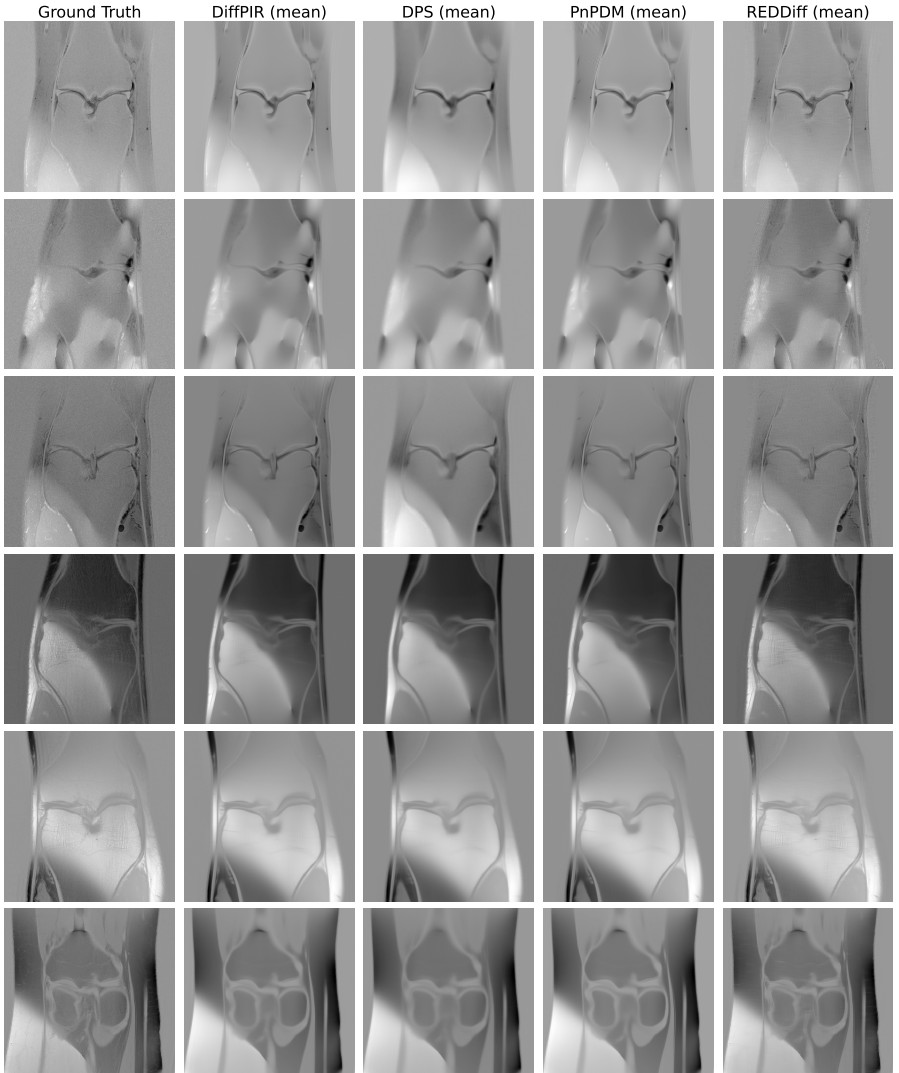

*Figure 12.* The mean of the posterior samples from the compressed sensing MRI inverse problem. Despite high fidelity, it does not directly mean the posteriors are well calibrated.

As shown in Table 2, both BF and Mira consistently assign higher scores to the models with less noise, with the ranking degrading smoothly as noise increases.

### E.2.2. GAUSSIAN MIXTURE MODEL SHIFTS

For this experiment, we consider a non-conditional high-dimensional setup where we define Gaussian Mixture Models (GMMs) of 100 dimensions and 20 mixture components. To consider different models, we create a distributional shift by translating the means along the diagonal direction, with the unshifted distribution being the true model.

To compute Mira, from each GMM, we generate $N = 5\,000$ samples, $L = 5\,000$ true samples, and consider 100 regions per true sample. As this experiment does not involve Bayesian inference, there is no likelihood or prior. Instead, we treat the GMM probability density function (pdf) as our evidence and compute the BF as the ratio of the candidate model pdf to the true model pdf. To make it consistent with Mira, we average the BF over the $L = 5\,000$ true samples.

As shown in Table 3, both Mira and the BF correctly identify the unshifted model ($\ell = 0$) as the most accurate and identify increasingly shifted models as less probable. The Mira score transitions from $2/3$ for the well-calibrated model toward the lower bound of $1/2$ as the shift increases, while the log BF becomes increasingly smaller.

*Table 3.* Comparison of Mira and BF scores across GMM shift magnitudes. Mira reliably favors the in-distribution model and penalizes shifted ones. Mira scores range from $1/2$ (poorly specified model) to $2/3$ (well specified model). BF near 1 indicate models that are nearly as plausible as the reference model $\mathcal{M}^*$; lower values indicate less support.

| Shift Magnitude | Mira | BF |
|---|---|---|
| -6 | 0.5040 | 0.000 |
| -3 | 0.5878 | $4.55 \times 10^{-145}$ |
| **0** | **0.6664** | **1.000** |
| 3 | 0.5925 | $5.45 \times 10^{-162}$ |
| 6 | 0.5034 | 0.000 |

Across both experiments, we observe strong agreement between Mira and the BF. This indicates that Mira provides a reliable, sample-based criterion for model comparison. Unlike the BF, which compares models via the marginal likelihood in data space, Mira operates directly on posterior samples in parameter space, making it a practical and complementary alternative for Bayesian model selection, particularly in settings where evidence computation is intractable.

### E.3. Comparison with Other Accuracy Scores

We aim here to benchmark the sensitivity of Mira against other existing conditional distribution calibration metrics.

E.3.1. COMPARISON WITH WITH MMD, WASSERSTEIN, C2ST ON BAYESIAN POSTERIOR ACCURACY EVALUATION

First, in the Bayesian inference setting, we compare the sensitivity of Mira against that of the two-sample tests C2ST, MMD, and the Wasserstein distance. We want to evaluate the capacity of these metrics to detect adversarial perturbations of posterior distributions.

We define the inference problem as a linear Gaussian model with prior $y \sim \mathcal{N}(0, I_2)$ and data distribution $x \mid y \sim \mathcal{N}(y, 0.5^2 I_2)$. For every perturbation considered, we draw $L = 200$ true samples $y^*$ from the prior distribution, and a corresponding observation $x^*$ for each. From this, we draw $N = 500$ posterior samples for each observation from the analytical posterior.

In addition to testing the scores on the samples from the true analytical posteriors as a baseline, we consider two perturbations: First, the heavy-tail perturbation, for which we replace the analytical Gaussian posterior by a location-scale Student's t distribution with mean set to match the true posterior and with two degrees of freedom. The second perturbation we consider is a 2-dimensional rotation of the posteriors by a fixed $40^o$ about the MAP. This transformation preserves all the pairwise distances between the posterior samples, but breaks the relation between the point cloud and the true sample $y^*$.

Results are displayed in Table 4. Under the null test, Mira, C2ST, MMD, and the Wasserstein distance correctly identify that the posterior estimators are accurate. In the case of the heavy-tailed perturbation, while Mira, C2ST, and the Wasserstein distance can detect the distortion, MMD shows limited sensitivity. In the case of the rotated posteriors, C2ST, MMD, and Wasserstein remain near their calibrated values due to the distance invariance of the perturbation. Mira, on the other hand, correctly identifies that the posteriors are not calibrated. This reveals that traditional scores of posterior calibration can be blind to some discrepancies in posterior estimators, especially when posteriors are not calibrated in ways that respect certain symmetries (such as distance invariance, or when marginals are identical to those of the unconditional distribution).

*Table 4.* Scores sensitivity comparison for different posterior perturbations (see text for details).

| Setting | Mira | C2ST | MMD | Wasserstein |
|---|---|---|---|---|
| Null | $0.6599 \pm 0.0155$ | $0.5271 \pm 0.0005$ | $0.0013 \pm 0.0008$ | $0.0689 \pm 0.0127$ |
| Heavy Tailed | $0.6215 \pm 0.0200$ | $0.7880 \pm 0.0005$ | $0.0194 \pm 0.0020$ | $0.3835 \pm 0.0308$ |
| Rotated | $0.6954 \pm 0.0152$ | $0.5934 \pm 0.0050$ | $0.0015 \pm 0.0009$ | $0.0707 \pm 0.0139$ |

E.3.2. COMPARISON WITH MMD, WASSERSTEIN, C2ST ON TAIL AND DIMENSION SENSITIVITY

Next, Tables 5 and 6 evaluate the sensitivity of Mira relative to MMD, the Wasserstein distance, and C2ST as the dimensionality of the $\mathcal{Y}$ space, $d_y$, varies. To simulate distributional shifts, we systematically attenuate the tail mass of the

conditional distribution: beyond a parametric cutoff, the density is scaled by a factor $\alpha < 1$ (specifically, we consider a cutoff of 4.0 with $\alpha = 0.8$ and a cutoff of 2.0 with $\alpha = 0.4$). This effectively supresses rare events without fully truncating them. Mira consistently detects these shifts, with scores scaling according to the degree of suppression, whereas standard metrics fail to flag them. Moreover, as shown here and in Appendix E.5 (Figure 16), Mira scales effectively with dimensionality. This reliability highlights Mira's utility for assessing conditional model calibration, particularly in settings where the modeling of rare tail events is critical but prone to under-estimation.

*Table 5.* Results in 2 dimensions for varying levels of tail suppression.

| Label | Mira | MMD | Wasserstein | C2ST |
|---|---|---|---|---|
| Null | $0.6628 \pm 0.0205$ | $0.00005 \pm 0.0002$ | $0.0429 \pm 0.0094$ | $0.5009 \pm 0.0094$ |
| cutoff=4.0, scale=0.80 | $0.6317 \pm 0.0243$ | $0.0001 \pm 0.0003$ | $0.0448 \pm 0.0107$ | $0.4998 \pm 0.0097$ |
| cutoff=2.0, scale=0.40 | $0.6027 \pm 0.0246$ | $0.0000 \pm 0.0002$ | $0.0430 \pm 0.0085$ | $0.5007 \pm 0.0084$ |

*Table 6.* Results in 100 dimensions for varying levels of tail suppression.

| Label | Mira | MMD | Wasserstein | C2ST |
|---|---|---|---|---|
| Null | $0.6568 \pm 0.0224$ | $0.000021 \pm 0.000037$ | $0.0430 \pm 0.0022$ | $0.4960 \pm 0.0097$ |
| cutoff=4.0, scale=0.80 | $0.644229 \pm 0.0250$ | $0.000023 \pm 0.000039$ | $0.0430 \pm 0.0019$ | $0.4950 \pm 0.0096$ |
| cutoff=2.0, scale=0.40 | $0.5756 \pm 0.0312$ | $0.000027 \pm 0.000032$ | $0.0428 \pm 0.0020$ | $0.4974 \pm 0.0095$ |

### E.3.3. COMPARISON WITH MMD AND WASSERSTEIN DISTANCE ON CONDITIONAL GENERATIVE MODEL EVALUATION

Next, we benchmark Mira against MMD (with radial basis function kernel) and the Wasserstein distance using the conditional generative models described in Section 5.2. For each of the 10 MNIST digits, draw $L = 100$ fiducial samples $y^*$ made of real MNIST test images and compare each against $N = 80$ samples drawn from the proposal model. We consider 3 models in this experiment: (1) a null setting (real vs. real), where the proposal samples are drawn from the remaining held-out test data to establish a baseline; (2) a conditional Variational Autoencoder (CVAE); and (3) a conditional Diffusion Model. Results are reported as the average across all 10 digits.

*Table 7.* Comparison of Mira, MMD, and Wasserstein Distance for conditional generative model evaluation. Mira correctly ranks models according to their level of accuracy. In contrast, MMD and Wasserstein yield similar scores for both the null test and misscalibrated models, indicating limited sensitivity.

| Experiment | Mira | MMD | Wasserstein Distance |
|---|---|---|---|
| Null Test | 0.6718 | 0.003885 | 2.413185 |
| Conditional VAE | 0.5716 | 0.021919 | 2.390693 |
| Conditional Diffusion Model | 0.6599 | 0.010114 | 2.449833 |

As reported in Table 7, Mira correctly characterizes the divergence of the models proposal from the true data distribution. It assigns high scores to the Null setting, but significantly lower scores to the CVAE, which is known to produce lower-quality samples. The Diffusion model, on the other hand, achieves scores near the theoretical $2/3$ benchmark (expected for a well-calibrated posterior), demonstrating its improved conditional accuracy. Conversely, MMD and the Wasserstein distance produce comparable values for both the Null and the samples from the CVAE model, indicating limited sensitivity to moderate distributional shifts in this low-sample regime.

### E.3.4. COMPARISON WITH L-C2ST ON BAYESIAN POSTERIOR ACCURACY EVALUATION

Next, we benchmark Mira against the Local Classifier Two-Sample Test (LC2ST; (Linhart et al., 2023)), a local posterior diagnostic that trains a binary classifier to distinguish samples drawn from the true posterior $p(y \mid x_o)$ versus the approximate posterior $q(y \mid x_o)$, with statistical significance assessed via a permutation-calibrated test statistic. While LC2ST operates within the same conditional comparison framework as Mira, it is fundamentally limited by the capacity of its learned classifier. This classifier can degrade significantly in high-dimensional settings, particularly when discrepancies appear

in higher-order structures of distributions like tail behavior rather than in low-order moments. To demonstrate this, we construct a 50-dimensional example where the true posterior is a factorized Laplace distribution (zero mean, unit variance) and the approximate posterior is a standard multivariate Gaussian $\mathcal{N}(0, I_{50})$. These distributions are matched in mean and variance, differing only in their tail structure. For each observation, we draw 100 samples from each posterior and compute the LC2ST statistics using a permutation-based null distribution. LC2ST yields near-null values (Table 8), failing to distinguish the heavy-tailed Laplace from the Gaussian. In contrast, Mira consistently identifies the mismatch. This result highlights that classifier-based two-sample tests can suffer severe power loss under high-dimensional distributional shifts, whereas Mira retains its sensitivity.

*Table 8.* Comparison of MIRA and LC2ST on a 50-dimensional Laplace distribution and a Gaussian distribution. MIRA successfully detects the mismatch, whereas LC2ST fails to reject the null.

| Quantity | Mira | LC2ST |
|---|---|---|
| Score | $0.6457 \pm 0.0251$ | $0.0660 \pm 0.0710$ |

### E.3.5. COMPARISON WITH SBC AND SLICED WASSERSTEIN DISTANCE ON CONDITIONAL MODEL EVALUATION

Finally, we compare the sensitivity of SBC, Mira, and the Sliced Wasserstein Distance (SWD) on a high-dimensional conditional generation task using MNIST. In each iteration, we draw a conditioning parameter $x = (p_0, p_2, p_4)$ from a Dirichlet distribution, representing class mixing proportions for digits 0, 2, and 4. A single reference image $y^*$ is generated by sampling a digit label according to $x$ and randomly selecting a corresponding image from the MNIST dataset, with small additive Gaussian noise. We compare the accuracy of three conditional models that generate samples given this $x$: Model A (Oracle) generates samples using the true conditioning parameter $x$; Model B (mildly misspecified) uses a perturbed $x$ that slightly underestimates class 2, with slightly elevated noise; and Model C (severely misspecified) ignores the conditioning $x$, generating samples from uniform proportions over all three classes, with strongly elevated noise.

In high-dimensional observation spaces such as images, axis-aligned marginal testing can be insensitive as individual pixel dimensions carry limited signal about joint distributional structure. A natural remedy is to replace pixel-aligned marginals with random 1D projections, which can better capture joint variations. We therefore evaluate SBC both in its standard marginal form and with random projections, and additionally compare against SWD, which is itself projection-based by construction, and Mira.

We assess each method through two complementary tests: the Kolmogorov-Smirnov (KS) $D$ statistic, which measures deviation from a reference distribution and can be used to rank models, and the associated $p$-value for hypothesis testing. For SBC (both marginal and projected variants), the ranks are tested against the uniform distribution. For SWD, per-projection distances are tested against those of the oracle (Model A). For projection-based methods, we vary the number of projections across $\{10, 50, 100\}$. With $n = 15{,}000$ pooled values per model, the KS critical value at $\alpha = 0.05$ is $D_{\text{critical}} \approx 0.016$. This means that models with KS-D below this threshold are statistically indistinguishable from the reference.

As shown in Table 9, Table 10, and Table 11, marginal SBC rejects all three models including the oracle ($p = 0.000$ throughout), and the KS-D values incorrectly rank Model B above the oracle. Projected SBC resolves this degeneracy (the oracle is no longer rejected), but still fails to reliably detect Model B, whose KS-D fluctuates around $D_{\text{critical}}$ across projection counts. SWD correctly orders all three models by KS-D, yet its $p$-values and KS-D for Model B remain below the detection threshold, leaving B statistically indistinguishable from A. Random projections thus improve upon axis-aligned marginals by avoiding the degeneracy of uninformative pixel dimensions, but they still decompose the high-dimensional comparison into 1D slices, each of which carries only a weak signal about joint distributional structure.

Mira is the only method that detects and correctly ranks all three models with clear, stable separation (A $\approx 0.68$, B $\approx 0.63$, C $\approx 0.50$), consistently across all runs and projection settings. Its ball-based regions operate natively in the full space, capturing joint distributional structure without projection or aggregation. Additionally, Mira scores are directly comparable across datasets of different sizes $L$ (expected value $2/3$, theoretical uncertainty $\sqrt{1/(18L)}$), unlike the KS-D statistic whose null distribution depends on sample size.

### E.4. Uninformative Estimator

A critical failure mode for posterior estimators is when $\hat{p}(y \mid x) = p(y)$. As demonstrated by Lemos et al. (2023), standard diagnostic tools like HPD regions often fail to detect this pathology. To evaluate Mira's sensitivity to an uninformative

*Table 9.* Comparison of SBC, SWD, and Mira with 10 projections.

| | SBC Marginal | | SBC Projected | | SWD vs A | | |
|---|---|---|---|---|---|---|---|
| Model | KS-D | $p$-val | KS-D | $p$-val | KS-D | $p$-val | Mira |
| Model A | 0.2409 | 0.0000 | 0.0227 | 0.4240 | 0.0000 | 1.0000 | $0.6789 \pm 0.0134$ |
| Model B | 0.1303 | 0.0000 | 0.0187 | 0.6728 | 0.0137 | 0.9421 | $0.6281 \pm 0.0196$ |
| Model C | 0.2549 | 0.0000 | 0.0353 | 0.0472 | 0.1283 | 0.0000 | $0.4994 \pm 0.0185$ |

*Table 10.* Comparison of SBC, SWD, and Mira with 50 projections.

| | SBC Marginal | | SBC Projected | | SWD vs A | | |
|---|---|---|---|---|---|---|---|
| Model | KS-D | $p$-val | KS-D | $p$-val | KS-D | $p$-val | Mira |
| Model A | 0.2483 | 0.0000 | 0.0062 | 0.9336 | 0.0000 | 1.0000 | $0.6746 \pm 0.0117$ |
| Model B | 0.1283 | 0.0000 | 0.0080 | 0.7200 | 0.0055 | 0.9748 | $0.6308 \pm 0.0130$ |
| Model C | 0.2507 | 0.0000 | 0.0229 | 0.0008 | 0.1247 | 0.0000 | $0.4946 \pm 0.0169$ |

posterior estimator, we replicate the experiment performed in Section 4.3 of Lemos et al. (2023). We consider a prior $p(y) = \mathcal{N}(0, 1)$, with observations generated via $x \sim \mathcal{N}(\theta, 0.1)$, and we set $p(y) = p(y \mid x)$.

We show in Table 12 that Mira is blind to this failure mode, as it gives a score of $2/3$. However, similarly to TARP, this issue can be solved by making the regions dependent on the true observation $x^*$, for instance by choosing centers as $c = x^* + \epsilon$, where $\epsilon \sim \mathcal{U}(-0.05, 0.05)$. With this new region construction, the Mira score drops to $0.54$, thus identifying the discrepancy between the true and proposal posteriors.

### E.5. Sensitivity Analysis

In this section, we conduct a comprehensive sensitivity analysis to characterize the behavior of Mira under variation of the following factors: (1) the dimensionality of the conditional distribution; (2) the number of regions associated with each true observation; (3) the number of conditional samples $N$; (4) the number of true samples $L$; (5) the choice of distance metric $d$ used to define regions; (6) the distribution $p(c)$ used to specify region centers; and the support of the distribution $p(c)$.

**(1) Dimensionality of the conditional distribution**   We use the toy GMM experiment introduced in Appendix E.2.2 to study the behavior of Mira under a fixed forward model configuration while increasing the dimensionality. Specifically, we retain 20 Gaussian components and consider dimensions $d \in \{2, 5, 10, 100\}$ for each distributional shift. The number of posterior samples and true samples are kept fixed at $N = 400$ and $L = 500$, and we use 100 different regions per true observation $L$. As shown in Figure 13, Mira consistently ranks the models correctly across all dimensions, with the well-specified model (no shift) maintaining a score close to $2/3$. We observe a gradual degradation of the Mira score as dimensionality increases, indicating increased sensitivity to the wrong model in higher-dimensional settings.

**(2) Number of regions**   We use the toy GMM experiment introduced in Appendix E.2.2, consisting of 20 Gaussian components in a 100-dimensional space. We fix the number of samples and true samples to $N = 400$ and $L = 500$, and vary the number of regions used per true observation $L$ to estimate the Mira score. We test with $\{1, 20, 50, 100\}$ regions. As shown in Figure 13, the estimated Mira score remains stable, indicating that the score is robust to the choice of the number of regions, even in moderately high-dimensional settings with a limited number of posterior samples.

*Table 11.* Comparison of SBC, SWD, and Mira with 100 projections.

| | SBC Marginal | | SBC Projected | | SWD vs A | | |
|---|---|---|---|---|---|---|---|
| Model | KS-D | $p$-val | KS-D | $p$-val | KS-D | $p$-val | Mira |
| Model A | 0.2448 | 0.0000 | 0.0047 | 0.8980 | 0.0000 | 1.0000 | $0.6782 \pm 0.0139$ |
| Model B | 0.1233 | 0.0000 | 0.0086 | 0.2197 | 0.0073 | 0.3991 | $0.6366 \pm 0.0133$ |
| Model C | 0.2532 | 0.0000 | 0.0224 | 0.0000 | 0.1298 | 0.0000 | $0.4976 \pm 0.0151$ |

*Table 12.* We report the mean and standard deviation for both single-run and bootstrap estimates. While random (unconditioned) centers fail to detect that the postererio estimator is not calibrated, x-dependent centers correctly identify that the model is conditionally independent of the data

| Center Selection Strategy ($c$) | Mira |
|---|---|
| Random (Unconditioned) | $0.6665 \pm 0.0071$ |
| x-Dependent (Conditioned) | $0.5412 \pm 0.0095$ |

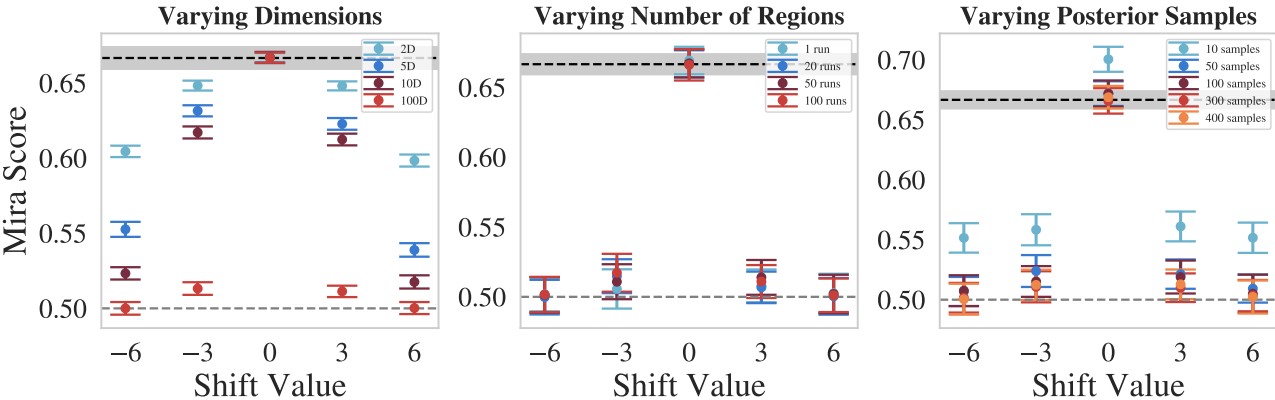

*Figure 13.* Mira score sensitivity under varying experimental conditions. Top-left: effect of dimension on score; Top-right: number of hyperspheres per fiducial; Bottom-left: number of posterior samples. Across settings, Mira scores peak for the well-calibrated ($\ell = 0$) model and fall to the poorly calibrated limit with increasing shift.

**(3) Number of conditional samples** $N$    We use the toy GMM experiment introduced in Appendix E.2.2, consisting of 20 Gaussian components in a 100-dimensional space. We fix the number of regions per true observation $L$ to 100 and vary the number of posterior samples as $N \in \{10, 50, 100, 300, 400\}$. As shown in Figure 13, Mira correctly ranks the models across all number of sample. However, with very few samples ($N = 10$), the estimate is noisy, while stable and accurate estimates emerge once $N = 100$.

**(4) Number of true samples** $L$    We use the linear regression experiment described in Appendix E.2.1 to study the effect of the number of true samples $L$ on the Mira score. We fix the number of posterior samples per model to $N = 5\,000$ and use 100 regions per true sample. The number of distinct ground-truth draws is varied from 10 to 1,000. The results, shown in Figure 14, demonstrate that Mira correctly ranks the models, where increasing noise leads to increasingly incorrect predictions, once $L \geq 50$. Moreover, as expected, when the number of true samples increases, the confidence intervals of Mira become narrower.

To further investigate the small-$L$ regime, we reproduce the strong lensing experiment of Section 5.3.2, a more challenging inference problem than the linear regression setting, with high-dimensional posteriors that can exhibit complex structure. The cost of simulation further restricts us to only $N = 64$ posterior samples per observation. This setting is therefore representative of realistic complex inference tasks, where both $L$ and $N$ are severely constrained. Apart from $L$, we use the same configuration (number of regions, distribution $p(c)$, etc.) as in Section 5.3.2. As shown in Table 13, even with a single fiducial sample Mira successfully identifies the best-performing model ($p_s(x)$, $\sigma_n = 2$), though it struggles to discriminate among poorly calibrated models. As $L$ increases, Mira recovers the correct model ranking.

**(5) Choice of distance metric**    To test Mira's sensitivity to the choice of distance metric used to build the random regions (see Equation 3), we use the linear regression experiment from Appendix E.2.1 and consider five distance metrics: L2 (Euclidean), L1 (Manhattan), Chebyshev, Cosine, and Minkowski (with $p = 3$). Each distance metric induces a different region geometry. Similarly to Appendix E.2.1, we use $L = 5\,000$ true samples and for each observation generate $N = 5\,000$ samples from the posterior. The number of regions per fiducial is fixed to 100. We show the results in Figure 15. All metrics successfully identify the least biased model (with $\eta = 0.001$) as the best-calibrated, and correctly rank the remaining models.

We further evaluate the choice of distance metric on high-dimensional image data using the strong lensing experiment of

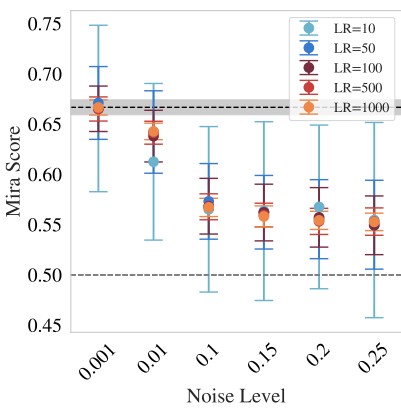

*Figure 14.* Mira score vs noise level for varying counts of true posteriors. As the number of true posteriors increases, confidence in the Mira estimate improves, and the distinction between well- and poorly-calibrated models becomes more pronounced.

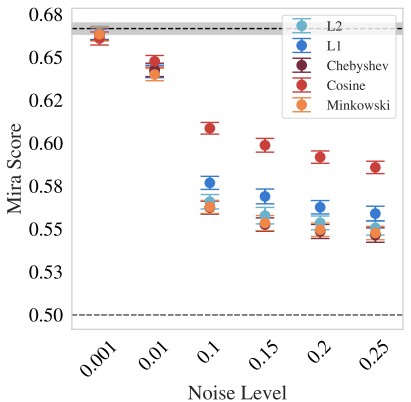

*Figure 15.* Mira scores for different distance metrics as a function of noise level.

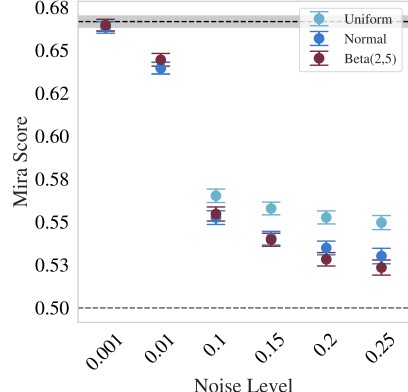

*Figure 16.* Mira scores for different center distributions as a function of noise level.

Section 5.3.2 with the same Mira configuration. As shown in Table 14, Mira produces nearly identical scores and consistent model rankings across L2, L1, Cosine, and Minkowski metrics. The exception is Chebyshev distance, which is known to become degenerate in high-dimensional spaces, rendering it unsuitable for $12\,288$-dimensional image vectors. Excluding this pathological case, Mira's rankings remain consistent across standard pixel-space metrics, further confirming its robustness to the choice of distance metric.

**(6) Center distribution** $p(c)$  To test Mira 's sensitivity to the choice of distribution used to sample centers from the label space (see Equation 3), we replicate the linear regression setup from Appendix E.2.1. We compare three center distributions: uniform distribution $\mathcal{U}[0, 1]$, standard normal distribution $\mathcal{N}(0, 1)$, and asymmetric Beta distribution $\mathcal{B}(2, 5)$. We use the same samples comfiguration as in Appendix E.2.1: $L = 5\,000$, $N = 5\,000$ and 100 regions. As shown in Figure 16, Mira scores rank the model accurately across all three distributions, highlighting Mira's robustness to the choice of center distribution.

**(7) Support of center distribution** $p(c)$  To test Mira's sensitivity to the support of the distribution $p(c)$, we repeat experiment of Section 5.1 with, $c \sim U[-10, 10]$ and $c \sim \mathcal{N}[-5, 1]$ and compare to the original $c \sim U[0, 1]$. We use the same configuration as in Section 5.1. Our results in Table 15 demonstrate that Mira is insensitive to this choice.

Overall, Mira remains robust across the sensitivity tests, in the sense that it consistently ranks the models correctly. The only exception occurs in the experiment with a varying number of true samples, in the extreme case where only 10 true samples are used. In this regime, the model with noise 0.2 is ranked slightly above the one with noise 0.15. This issue is resolved by adding 40 additional simulations to the setup. These results indicate that Mira is well-suited for practical use in simulation-based tasks under realistic computational constraints.

Table 13. Effect of the number of ground truths on Mira scores.

| # Ground Truths | $p_s(x)$, $\sigma_n = 2$ | $p_s(x)$, $\sigma_n = 0.5$ | $p_e(x)$, $\sigma_n = 2$ | $p_e(x)$, $\sigma_n = 0.5$ |
|---|---|---|---|---|
| 1 | $0.6722 \pm 0.2508$ | $0.5242 \pm 0.2919$ | $0.5136 \pm 0.2969$ | $0.5452 \pm 0.2715$ |
| 2 | $0.7042 \pm 0.1426$ | $0.5037 \pm 0.2033$ | $0.4844 \pm 0.2195$ | $0.5277 \pm 0.2071$ |
| 3 | $0.6522 \pm 0.1321$ | $0.5502 \pm 0.1596$ | $0.5065 \pm 0.1858$ | $0.5348 \pm 0.1466$ |
| 4 | $0.6190 \pm 0.1246$ | $0.5521 \pm 0.1362$ | $0.4998 \pm 0.1373$ | $0.5056 \pm 0.1459$ |
| 5 | $0.6642 \pm 0.1025$ | $0.5508 \pm 0.1235$ | $0.5025 \pm 0.1463$ | $0.5163 \pm 0.1219$ |
| 6 | $0.6213 \pm 0.0974$ | $0.5659 \pm 0.1095$ | $0.4968 \pm 0.1195$ | $0.5143 \pm 0.1268$ |
| 7 | $0.6448 \pm 0.0958$ | $0.5475 \pm 0.1030$ | $0.4939 \pm 0.1096$ | $0.5011 \pm 0.1058$ |
| 8 | $0.6501 \pm 0.0774$ | $0.5693 \pm 0.0880$ | $0.5132 \pm 0.0974$ | $0.5217 \pm 0.1055$ |
| 9 | $0.6615 \pm 0.0855$ | $0.5780 \pm 0.0906$ | $0.5133 \pm 0.0997$ | $0.5195 \pm 0.0988$ |
| 10 | $0.6416 \pm 0.0879$ | $0.5780 \pm 0.0942$ | $0.4878 \pm 0.0830$ | $0.5117 \pm 0.0850$ |
| 11 | $0.6417 \pm 0.0693$ | $0.5593 \pm 0.0893$ | $0.5047 \pm 0.1005$ | $0.5019 \pm 0.0822$ |
| 12 | $0.6555 \pm 0.0692$ | $0.5721 \pm 0.0770$ | $0.5020 \pm 0.0907$ | $0.5046 \pm 0.0961$ |
| 13 | $0.6685 \pm 0.0674$ | $0.5769 \pm 0.0743$ | $0.5025 \pm 0.0755$ | $0.5217 \pm 0.0732$ |
| 14 | $0.6558 \pm 0.0652$ | $0.5818 \pm 0.0660$ | $0.5331 \pm 0.0799$ | $0.5058 \pm 0.0828$ |
| 15 | $0.6465 \pm 0.0628$ | $0.5795 \pm 0.0674$ | $0.5249 \pm 0.0685$ | $0.5194 \pm 0.0676$ |
| 16 | $0.6484 \pm 0.0595$ | $0.5774 \pm 0.0715$ | $0.5259 \pm 0.0692$ | $0.5026 \pm 0.0701$ |

Table 14. Effect of distance metric on Mira scores.

| Metric | $p_s(x)$, $\sigma_n = 2$ | $p_s(x)$, $\sigma_n = 0.5$ | $p_e(x)$, $\sigma_n = 2$ | $p_e(x)$, $\sigma_n = 0.5$ |
|---|---|---|---|---|
| L2 | $0.6489 \pm 0.0576$ | $0.5722 \pm 0.0651$ | $0.5281 \pm 0.0793$ | $0.5002 \pm 0.0721$ |
| L1 | $0.6455 \pm 0.0601$ | $0.5800 \pm 0.0593$ | $0.5263 \pm 0.0667$ | $0.5020 \pm 0.0670$ |
| Chebyshev | $0.6670 \pm 0.0592$ | $0.5122 \pm 0.0592$ | $0.5378 \pm 0.0764$ | $0.5959 \pm 0.0669$ |
| Cosine | $0.6490 \pm 0.0564$ | $0.5418 \pm 0.0665$ | $0.5105 \pm 0.0833$ | $0.5293 \pm 0.0709$ |
| Minkowski ($p = 3$) | $0.6476 \pm 0.0625$ | $0.5734 \pm 0.0754$ | $0.5202 \pm 0.0712$ | $0.5068 \pm 0.0761$ |

Table 15. Comparison of different center distributions support.

| Center Distribution | Correct | Overconfident | Underconfident | Biased |
|---|---|---|---|---|
| $\mathcal{U}[0, 1]$ | $0.6677 \pm 0.0072$ | $0.6144 \pm 0.0079$ | $0.6937 \pm 0.0073$ | $0.5448 \pm 0.0089$ |
| $\mathcal{U}[-10, 10]$ | $0.6700 \pm 0.0066$ | $0.6150 \pm 0.0081$ | $0.6941 \pm 0.0061$ | $0.5448 \pm 0.0095$ |
| $\mathcal{N}[-5, 1]$ | $0.6671 \pm 0.0074$ | $0.6162 \pm 0.0081$ | $0.6930 \pm 0.0057$ | $0.5480 \pm 0.0090$ |

