# OpenReview forum: "MIRA: A Score for Conditional Distribution Accuracy and Model Comparison"
_ICML.cc/2026/Conference — ICML 2026 spotlight_

### Official Review · Reviewer_Ychm · 2026-03-02

**Soundness:** 2
**Presentation:** 2
**Significance:** 3
**Originality:** 2
**Overall Recommendation:** 4
**Confidence:** 4

**Summary:**

The paper proposes a new method to check the uncertainty calibration of (conditional) generative models and perform model comparison. It is based on comparing the distances of true and model data from random query points and might thus be sensitive to other failure modes than existing tests. This is a valuable addition to the model validation toolbox.

**Compliance With Llm Reviewing Policy:**

Affirmed.

**Final Justification:**

The additional experiment, along with other proposed changes, considerably improved the paper. I've raised my score to "4: weak accept".

**Key Questions For Authors:**

See above.

**Limitations:**

While the method in itself is sound, comparisons with the state-of-the-art are missing.

**Strengths And Weaknesses:**

The paper proposes the MIRA ("mass in random areas") statistic, derives its properties under the null hypothesis "true and model distributions are equal", defines the MIRA test, designs practical estimation algorithms, and conducts experiments to demonstrate the properties of the MIRA test. This may become a valuable method in the validation toolbox, but is not yet sufficiently mature in its current form.

First of all, it is presented as a novel idea that considerably deviates from existing validation practice, but on closer inspection this turns out to be incorrect. In fact, MIRA is just a new variant of the well-established Probability Integral Transform (PIT) methodology. This can be seen by rephrasing the essence of the proposed algorithm as follows:
1. Given a GT pair (x*, y*), sample a set of model predictions {y_i ~ p(y | x*)} for i = 1...K.
2. With a random query point c ~ q(c), calculate d* = d(c, y*) and d_i = d(c, y_i).
3. Determine the order statistic of d* among the d_i.
4. Repeat this for many GT pairs. Under the null hypothesis, the order statistics d* must be ~ uniform(0, 1).

This is exactly the same algorithm as in standard PIT, except that the usual marginal ordering is replaced with distance ordering. Recognizing this generalization would allow the MIRA method to be presented much more elegantly. The current proofs of its properties appear grossly over-engineered in comparison.

Another novelty of the paper is the method to reduce the detailed calibration results to a single scalar, the MIRA score. It amounts to picking a random rank (by way of the distance d_r = d(c, y_r) for a random y_r ~ p(y | x*)) and determining if d* (i.e. the target order statistic) is below or above d_r (a random rank ~ uniform(0,1)). These decisions are aggregated into a weighted sum whose expectation and variance under the null are 2/3 and 1/18 respectively. This contrasts with standard PIT, where the aggregation into a scalar is usually performed with theoretically well-grounded Kolmogorov-Smirnov or Cramer-von Mises statistics over the empirical cumulative distribution function (ECDF). Moreover, aggregation in PIT is usually accompanied by histograms or ECDF plots, which show the detailed behavior of the order statistics. The coverage plots in the paper resemble this, but are non-standard and therefore harder to interpret.

In order for the paper to become publishable, the following are necessary:
* Properly embed MIRA in the existing work on model validation, especially PIT. At present, the "Related Work" is merely a collection of existing methods that are (judging by the lack of any comparison) claimed to be unrelated to the new proposal, but this is incorrect. The authors should analyze the prior art more carefully.
* Simplify the derivations and justifications in light of the rank-order interpretation described above.
* Define baselines from prior work and properly compare the new method against them. Analyse in detail for which failure modes of the learning process MIRA is more or less sensitive than existing methods. Determine if MIRA is competitive with existing model comparison strategies. Adopt established experimental protocols and benchmark problems from the literature.
* Compare the MIRA score to alternative statistical tests (Kolmogorov-Smirnov, Cramer-von Mises etc.) both in theory and empirically.
* Analyse in detail the effect of MIRA's hyperparameters, especially with regard to the query points c. What are the consequences of different choices of the distribution q(c)? How many query points are needed? What distance function d(c, y) should be used? How do these decisions effect the types of failures MIRA can or cannot detect?

---

> ### Author Rebuttal · Authors · 2026-03-31
>
> We thank the reviewer for this detailed feedback. We address each point below.
>
> **On the relationship with PIT.** We agree that Mira can be interpreted through the lens of rank statistics. However, we believe our contributions go beyond proposing a new PIT variant, in the following ways:
>
> 1. We deliberately framed the derivations in explicit Bayesian language, computing the posterior probability $p(k \mid n)$ via Laplace's rule of succession, rather than in the frequentist framework typical of PIT tests. This was a deliberate choice to connect Mira to recent methods gaining traction in the high-dimensional inverse problem community, such as HPD region coverage tests (Hermans et al., 2022), TARP (Lemos et al., 2023), and PQMass (Lemos et al., 2025). That said, we agree that the propositions we derive (closed-form statistic, theoretical mean 2/3, variance 1/18, and the new lower bound of 1/2) could equally be stated in the language of ranks. We have done so in the sufficiency proof for the modified score above, and we are happy to include such a reformulation in an appendix if the reviewer finds it valuable. We will also strengthen the related work section to make the connection with PIT more explicit.
> 2. Our main contribution is a specific algorithm that can rank models and quantify accuracy of conditional distributions with high sensitivity even in very high-dimensional settings. We compare Mira against methods from several methodological families. Against the PIT-based method SBC (Appendix B.6.5, Table 9), SBC fails to reliably rank misspecified MNIST conditional models, while Mira correctly orders all three. We also compare against methods from other families: the classifier-based LC2ST (Appendix B.6.4–B.6.5, Tables 8–9) fails to detect mismatch between a 50-dimensional Laplace and Gaussian distribution and yields near-null statistics across all MNIST conditions; the  MMD and Wasserstein distance (Appendix B.6.1–B.6.3, Tables 4–7) fail to detect rotated posteriors and tail perturbations that Mira identifies. Finally, in high-dimensional inverse problems (Section 5.3.2, Appendix B.4), Mira scales to $3 \times 64 \times 64$ dimensions with only 16 true samples and correctly ranks models on realistic scientific tasks from InverseBench (black hole imaging, compressed sensing MRI).
>
> **On sensitivity analysis and hyperparameters.** Appendix C provides a comprehensive sensitivity analysis covering all the factors the reviewer mentions: dimensionality (2D to 100D), number of regions (1 to 100), number of posterior samples $N$ (10 to 400), number of true samples $L$ (10 to 1000), five different distance metrics (L2, L1, Chebyshev, Cosine, Minkowski), and three different center distributions (Uniform, Normal, Beta(2,5)). Mira remains robust across all settings.
>
> We believe the current experiments provide extensive comparisons across multiple methodological families. If the reviewer has specific additional benchmarks or experimental protocols in mind that would provide further insight, we would be happy to include them.

---

> > ### Author Rebuttal · Reviewer_Ychm · 2026-04-02
> >
> > 1. I appreciate your commitment to a Bayesian framing; however, the distinction between Laplace’s Rule and the PIT/Rank-order framework is a matter of language rather than substance. For example, when the Agnesti-Coull adjustment is applied to PIT (as is quite common), the distinction reduces to different choices of prior: Beta(1,1) in Laplace’s Rule of Succession and Beta(2, 2) in PIT with Agnesti-Coull. Both are effectively a smoothing of the empirical CDF, and the latter is no less Bayesian than the former. By acknowledging the rank-order interpretation, you could provide much more intuitive geometric proofs, while still arriving at the same Bayesian posteriors. This would significantly increase the paper's impact among practitioners who are familiar with PIT, without sacrificing the Bayesian integrity of your derivations. It would also unlock a wealth of established tools from SBC (e.g. Kolmogorov-Smirnov and Cramer-von Mises statistics, ECDF plots) to gain additional insight about the precise nature of a detected mismatch, beyond a single numerical score.
> >
> > 2. Given the conceptual similarity between MIRA and SBC, I'm curious why the latter fails when the former succeeds (Appendix B.6.5, Table 9). Is it merely because SBC (as applied in this experiment) is restricted to marginal distributions, whereas MIRA uses anchor point distances? This would indicate that the crucial difference is not the choice of statistical approach, but the power of anchor point distances in comparison to marginals. For example, how does SBC fare when axis-aligned marginals are replaced with random projections, as in sliced Wasserstein distances?

---

> > > ### Author Response · Authors · 2026-04-07
> > >
> > > We thank the reviewer for their follow-up and experiment idea.
> > >
> > > ### On point 1
> > >
> > > We agree that the distinction between Laplace's Rule and PIT with Agresti-Coull reduces to a choice of prior, and we will make this connection explicit in the related work section. In addition to the rank-based sufficiency proof provided in response to Reviewer 9z8D, we will include a rank-order reformulation in an appendix to make the method more accessible to practitioners familiar with PIT.
> > >
> > > As the reviewer notes, this does not change the underlying theory, but we agree it improves accessibility for practitioners familiar with SBC and PIT.
> > >
> > > We also appreciate the suggestion regarding KS and Cramér–von Mises statistics. We view this as a promising direction for future work and will discuss this in the revision.
> > >
> > > ### On point 2
> > >
> > > We thank the reviewer for this excellent question, which prompted us to revisit Experiment B.6.5. In the original submission, SBC was applied after embedding images into a 3-dimensional space, which assumed access to an informative embedding that may not be available in practice. We have repeated the experiment directly in 784-dimensional pixel space, extending the comparison to include marginal SBC (axis-aligned), projected SBC (random projections), and the Sliced Wasserstein Distance (SWD), varying the number of projections across $\{10, 50, 100\}$. For each method, we evaluate model ranking using both the KS $D$ statistic and its associated $p$-value. SBC ranks are tested against the uniform distribution; SWD distances are tested against Model A’s (the oracle). With $n = 15{,}000$ pooled values per model, the KS critical value at $\alpha = 0.05$ is $D_{\mathrm{critical}} = 1.36\sqrt{2/15{,}000} \approx 0.016$: models with KS-D below this threshold are statistically indistinguishable from the null.
> > >
> > > **Marginal SBC** (axis-aligned) rejects all three models, including the oracle ($p = 0.000$ for all). The KS-D values incorrectly rank Model B as better calibrated than the oracle ($0.13$ vs $0.24$). Both the $p$-value and KS-D are thus unreliable: the $p$-value rejects correctly specified models, and the KS-D ranking is wrong.
> > >
> > > **Projected SBC** (random projections) resolves the marginal degeneracy: the oracle is no longer rejected ($p > 0.05$). However, it still fails to detect Model B reliably: the $p$-value for B never drops below $0.05$ at any projection count, and the KS-D for B fluctuates between $0.005$ and $0.019$, remaining close to $D_{\mathrm{critical}} \approx 0.016$. Only Model C is consistently detected at 50+ projections.
> > >
> > > **SWD** correctly ranks models (A $<$ B $<$ C), but Model B remains indistinguishable from A: $p$-values stay above $0.05$, and KS-D stays below $D_{\mathrm{critical}}$. Only Model C is detected.
> > >
> > > **Mira** is the only method that detects and correctly ranks all three models with clear, stable separation (A $\approx 0.68$, B $\approx 0.63$, C $\approx 0.50$), consistently across all runs and settings.
> > >
> > > **10 projections**
> > > | Model | SBC Marg KS-D | Marg p-val | SBC Proj KS-D | Proj p-val | SWD KS-D vs A | SWD p-val | MIRA |
> > > | -------- | ------------- |-------------| ------------- |------------- |------------- |-------------| ------------- |
> > > | Model A  | 0.2409 | 0.0000  | 0.0227| 0.4240| 0.0000| 1.0000| 0.6789 ± 0.0134 |
> > > | Model B  | 0.1303 | 0.0000  | 0.0187| 0.6728| 0.0137| 0.9421| 0.6281 ± 0.0196 |
> > > | Model C  | 0.2549 | 0.0000  | 0.0353| 0.0472| 0.1283| 0.0000| 0.4994 ± 0.0185 |
> > >
> > > **50 projections**
> > > | Model   | SBC Marg KS-D  | Marg p-val | SBC Proj KS-D  | Proj p-val | SWD KS-D vs A  | SWD p-val  | MIRA
> > > | -------- | ------------- |-------------| ------------- |------------- |------------- |-------------| ------------- |
> > > Model A | 0.2483| 0.0000| 0.0062| 0.9336| 0.0000| 1.0000| 0.6746 ± 0.0117 |
> > > Model B | 0.1283| 0.0000| 0.0080| 0.7200| 0.0055| 0.9748| 0.6308 ± 0.0130
> > > Model C | 0.2507| 0.0000| 0.0229| 0.0008| 0.1247| 0.0000| 0.4946 ± 0.0169
> > >
> > > **100 projections**
> > > Model   | SBC Marg KS-D  | Marg p-val| SBC Proj KS-D| Proj p-val| SWD KS-D vs A| SWD p-val | MIRA|
> > > | -------- | ------------- |-------------| ------------- |------------- |------------- |-------------| ------------- |
> > > Model A | 0.2448| 0.0000| 0.0047| 0.8980| 0.0000| 1.0000| 0.6782 ± 0.0139
> > > Model B | 0.1233| 0.0000| 0.0086| 0.2197| 0.0073| 0.3991| 0.6366 ± 0.0133
> > > Model C | 0.2532| 0.0000| 0.0224| 0.0000| 0.1298| 0.0000| 0.4976 ± 0.0151
> > >
> > > In summary, random projections improve upon axis-aligned marginals by avoiding the degeneracy of uninformative pixel dimensions. However, they still decompose the high-dimensional comparison into 1D slices, each of which carries a weak signal. Mira’s ball-based regions operate natively in the full space, capturing joint distributional structure without projection or aggregation. Additionally, Mira scores are directly comparable across datasets of different sizes $L$ (expected value $2/3$, uncertainty $\sqrt{1/(18L)}$), unlike the KS $D$.

---

### Official Review · Reviewer_juwu · 2026-03-05

**Soundness:** 3
**Presentation:** 3
**Significance:** 4
**Originality:** 4
**Overall Recommendation:** 5
**Confidence:** 3

**Summary:**

This article derives a sample-based score called MIRA, a score that allows assessing the faithfullness of a conditional candidate distribution of model $M: p(y | x, M)$, to that of a true probabilistic model $M*: p(y | x, M*)$. The score is simple to compute and relies on L samples from the true distribution $(x*, y*) \sim p(x, y | M*)$, and for each sample x*, N samples from the candidate distribution $y_j ~ p(y | x*, M)$.

Over a course of several examples, the authors demonstrate its usefulness in several contexts. Namely, the score can be used for identifying unfaithful posterior approximators, and as such provide an alternative to model diagnostic tools such as TARP and SBC. Second, the authors show its use in a Bayesian model comparison context, offering an alternative to metrics such as Bayes factors.

**Compliance With Llm Reviewing Policy:**

Affirmed.

**Key Questions For Authors:**

1. The case studies demonstrated that the method works with as little samples as L=16 and N=64. How much further down is it possible to push the requirements down? Would the method work well with L=1?

2. Is there a clear interpretation of differences between MIRA scores? e.g., is a model with score ~0.7 about the same quality as model with score ~0.62 (both being about the same distance from 2/3)?

3. By default, the center of the region is drawn from a uniform distribution. Some work has been done to show robustness against this choice with two alternative distributions (normal and beta). However, in all cases, it appears that the proposal distribution always has a substantial overlap with the both conditional distributions. What happens if the approximate or true distribution is much different from the proposal? e.g., if the posterior is very narrow, one would end up drawing many centers far from the distribution. How would this affect the convergence rate of the score?

**Limitations:**

Yes.

**Strengths And Weaknesses:**

The main feature of the method lies in the assumption of having an access to L samples from the true joint distribution $(x*, y*) \sim p(x, y | M*)$ and assessing the "closeness" of the candidate conditional $M: p(y | x*, M)$ to that of $M*: p(y | x*, M*)$. The resulting score is a scalar bounded between 0 and 1, with a clear description of its distributional properties under the null (that the two conditionals are identical). The proposed score is therefore clearly very useful in the context of simulation-based inference (SBI), as it allows to determine whether a given posterior approximator faithfully captures its true (typically unacessible) counterpart. As it provides a numerical summary, it sets itself apart from TARP and SBC plots which need to be evaluated visually. Compared to SBC and its numerical summary the log-gamma statistic (see Modrák, et al., 2025 for new extensions of SBC), it further assesses the joint distribution of $y$ rather than its marginals. This makes it particularly appealing in high-dimensional settings with complicated dependency structures.

The downside of the metric for SBI is that *because* it provides a single numerical summary for the whole conditional distribution, it may tell us *whether* or *how much* the candidate model is unfaithful, but it does not provide qualitative assessment in *what way* it is misspecified. Nevertheless, for its advantages, the MIRA score might become one of the default diagnostic metrics of computational faithfulness of SBI models as it addressess weaknesses of other methods.

It is the application of the MIRA score to Bayesian model comparison where the limitations become clearer. First, MIRA score requires L samples from the true joint distribution. However, in many realistic use cases of Bayesian model comparison, all we have is samples from the true marginal $x* \sim p(x | M*)$ without ground truth labels. Further, in many applications L=1 in model comparison settings. Second, the method seems to be applicable only in scenarios where all models have the same parameters; this is commonly not the case for many scientific models. Third, the authors mention that compared to Bayes factors, MIRA is clearly interpretable. The Bayes factor: a ratio of the probability of the data x* under one model over another model is clearly easy to interpret, and is further easy to convert into posterior probability over models. This is not the case for MIRA, where relative distances between MIRA scores are not easily translatable in the same way.

References

Modrák, M., Moon, A. H., Kim, S., Bürkner, P., Huurre, N., Faltejsková, K., ... & Vehtari, A. (2023). Simulation-based calibration checking for Bayesian computation: The choice of test quantities shapes sensitivity. Bayesian Analysis, 20(2), 461.

---

> ### Author Rebuttal · Authors · 2026-03-31
>
> We thank the reviewer for their insightful comments.
>
> ### Interpretability
> We show that the Mira score can provide interpretability in terms of over- and underconfidence.
>
> Starting from Eq 38, with $\mathbb{E}[\lambda_n] = 1/2$ (Prop 3.1) and recognizing that $2\,\mathbb{E}[\lambda_n\,\lambda_k] - \mathbb{E}[\lambda_k] = 2\mathrm{Cov}(\lambda_n, \lambda_k)$:
> $$
> \mu_{\mathrm{Mira}}(\mathcal{M}) \overset{N\to\infty}{\longrightarrow} \frac{1}{2} + 2\,\mathrm{Cov}(\lambda_n, \lambda_k).
> $$
> Under the null hypothesis ($\lambda_k = \lambda_n$ a.s.), this gives $\mu = 1/2 + 2\,\mathrm{Var}(\lambda_n) = 2/3$ (Cor 3.4). Deviations from $2/3$ are thus governed by how $\mathrm{Cov}(\lambda_n,\lambda_k)$ compares to $\mathrm{Var}(\lambda_n)$.
>
> Since $\mathrm{Var}(\lambda_n)=1/12$ is fixed (Prop 3.1), the key quantity is $\mathrm{Var}(\lambda_k)$: how much the true mass varies across random regions. An overconfident candidate concentrates its regions in a limited area of the true distribution, so $\lambda_k$ sees only a restricted range of true mass values, yielding $\mathrm{Var}(\lambda_k) < \mathrm{Var}(\lambda_n)$. Conversely, an underconfident candidate probes the full extent of the truth, and $\lambda_k$ swings from 0 to 1, yielding $\mathrm{Var}(\lambda_k) > \mathrm{Var}(\lambda_n)$.
>
>
> When $\mathrm{Var}(\lambda_k)<\mathrm{Var}(\lambda_n)$, By Cauchy Schwarz we derive $\mathrm{Cov}(\lambda_n,\lambda_k) < \mathrm{Var}(\lambda_n)$, hence $\mu < 2/3$. This motivates defining:
>
> - **Overconfident:** $\mathrm{Var}(\lambda_k) < \mathrm{Var}(\lambda_n) => \mu_{\mathrm{Mira}} < 2/3$ (by Cauchy Schwarz).
> - **Underconfident:** $\mu_{\mathrm{Mira}} > 2/3 => \mathrm{Var}(\lambda_k) > \mathrm{Var}(\lambda_n)$ (contrapositive).
>
>
> ### Small L
>
> We repeated Sec 5.3.2 starting with a single fiducial sample and 64 posterior samples, incrementing up to 16. Even with a single sample, Mira can detect the best model ($p_s(x),\sigma_n=2$) but strugle to differentiate poorly calibrated models. As the number of fiducial samples increases, Mira can rank models accurately.
>
> |# Ground Truths|$p_s(x),\sigma_n=2$|$p_s(x),\sigma_n=0.5$|$p_e(x),\sigma_n = 2$|$p_e(x), \sigma_n=0.5$|
> |-|-|-|-|-|
> |1|0.6722±0.2508|0.5242±0.2919|0.5136±0.2969|0.5452±0.2715|
> |2|0.7042±0.1426|0.5037±0.2033|0.4844±0.2195|0.5277±0.2071|
> |3|0.6522±0.1321|0.5502±0.1596|0.5065±0.1858|0.5348±0.1466|
> |4|0.6190±0.1246|0.5521±0.1362|0.4998±0.1373|0.5056±0.1459|
> |5|0.6642±0.1025|0.5508±0.1235|0.5025±0.1463|0.5163±0.1219|
> |6|0.6213±0.0974|0.5659±0.1095|0.4968±0.1195|0.5143±0.1268|
> |7|0.6448±0.0958|0.5475±0.1030|0.4939±0.1096|0.5011±0.1058|
> |8|0.6501±0.0774|0.5693±0.0880|0.5132±0.0974|0.5217±0.1055|
> |9|0.6615±0.0855|0.5780±0.0906|0.5133±0.0997|0.5195±0.0988|
> |10|0.6416±0.0879|0.5780±0.0942|0.4878±0.0830|0.5117±0.0850|
> |11|0.6417±0.0693|0.5593±0.0893|0.5047±0.1005|0.5019±0.0822|
> |12|0.6555±0.0692|0.5721±0.0770|0.5020±0.0907|0.5046±0.0961|
> |13|0.6685±0.0674|0.5769±0.0743|0.5025±0.0755|0.5217±0.0732|
> |14|0.6558±0.0652|0.5818±0.0660|0.5331±0.0799|0.5058±0.0828|
> |15|0.6465±0.0628|0.5795±0.0674|0.5249±0.0685|0.5194±0.0676|
> |16|0.6484±0.0595|0.5774±0.0715|0.5259±0.0692|0.5026±0.0701|
>
> ### Bayes Factor
>
> Regarding the interpretability of the BF, our previous wording was unclear; we meant to emphasize that the BF can be challenging to compute, especially in high dimensions. This will be corrected in the updated version. Mira should be seen as a diagnostic of model miscalibration complementary to the BF, not a replacement. This is discussed in more details in Appendix B.6.
>
> ### Choice of centers distribution
> We first note that Mira does not depend on $p(c)$ as long as $c$ has support where the distributions to be tested have support. By construction, every iteration constructs a region covering part of the mass of the proposal distribution.
>     Empirically, the choice of center distribution does not affect the sensitivity of Mira. We show this by repeating Experiment 5.1 with a broad proposal $c \sim U[-10, 10]$ and $c \sim \mathcal{N}[-5, 1]$ and compare to the original $c \sim U[0, 1]$. Our results demonstrate that Mira is insensitive to this choice.
>
> |Center Distribution|Correct|Overconfident|Underconfident|Biased|
> |-|-|-|-|-|
> |U[0, 1]|0.6677±0.0072|0.6144±0.0079|0.6937±0.0073|0.5448±0.0089|
> |U[-10, 10]|0.6700±0.0066|0.6150±0.0081|0.6941±0.0061|0.5448±0.0095|
> |$\mathcal{N}$[-5, 1]|0.6671±0.0074|0.6162±0.0081|0.6930±0.0057|0.5480±0.0090|

---

> > ### Author Rebuttal · Reviewer_juwu · 2026-04-01
> >
> > The authors addressed my questions. Thank you.

---

### Official Review · Reviewer_VNeR · 2026-03-09

**Soundness:** 4
**Presentation:** 3
**Significance:** 3
**Originality:** 3
**Overall Recommendation:** 5
**Confidence:** 4

**Summary:**

This paper introduces Mira (Mass In Random Areas), a sample-based score for assessing the accuracy of candidate conditional distributions using only joint samples from the true data-generating process. The method constructs random regions centered on draws from a reference distribution, counts how many candidate samples fall within each region, and then computes a Bayesian probability that the true sample also lies in that region. By averaging this statistic over observations and regions, the authors obtain a scalar score with known theoretical reference values: 2/3 for a correctly specified model and 1/2 for an independent (uninformative) model. They prove that the Mira statistic converges to a Beta(2,1) distribution under the null hypothesis, providing calibrated uncertainty estimates. Experiments span toy problems, MNIST conditional generation, and Bayesian inference tasks in astrophysics including gravitational lensing model comparison.

**Compliance With Llm Reviewing Policy:**

Affirmed.

**Key Questions For Authors:**

1. Mira provides a necessary but not sufficient condition for accuracy. Can you construct or identify a concrete example where two meaningfully different conditional distributions receive the same Mira score? This would help practitioners understand the method's blind spots.
2. The sensitivity analysis in Appendix C shows robustness to design choices, but all experiments use Euclidean distance. How sensitive is Mira to the choice of distance metric in structured output spaces (e.g., images), and would perceptual distances lead to meaningfully different conclusions?

**Limitations:**

yes

**Strengths And Weaknesses:**

**Soundness.** The theoretical development is solid and well-grounded. The key insight that the random region construction guarantees uniform mass under the candidate distribution (Proposition 3.1) is elegant and enables the closed-form Mira statistic via Laplace's rule of succession. The convergence to Beta(2,1) and the resulting theoretical mean and variance provide useful calibration references. However, Mira only provides a necessary condition for distributional accuracy, not a sufficient one: two distributions can assign equal mass to many random regions while still differing in structure. The authors acknowledge this limitation but do not characterize how severe it might be in practice, for instance whether there are natural distribution families where Mira is insensitive to important misspecifications.

**Presentation.** The paper is clearly written and well structured, with intuitive figures and a logical progression from theory to experiments.

**Significance.** The practical contribution is meaningful. Mira fills a genuine gap by providing a scalar, interpretable score for conditional distribution accuracy that works with only joint samples and scales to high dimensions. The comparison with TARP is particularly convincing: in the gravitational lensing experiments, TARP coverage curves are visually difficult to rank across models, while Mira provides an unambiguous ordering. The method is also computationally lightweight and does not require density evaluation or neural network training. That said, the experimental scope is somewhat narrow, focused on Bayesian inference in astrophysics and a single conditional generation task (MNIST). Demonstrating Mira on larger-scale generative modeling benchmarks would strengthen the case for broad adoption.

**Originality.** The approach builds naturally on PQMass (Lemos et al., 2025), adapting the idea of comparing probability mass in random regions from a frequentist two-sample setting to a Bayesian one-sample setting for conditional distributions. The Bayesian reformulation via Laplace's rule of succession is the key technical novelty that enables working with a single true sample per conditioning variable. The resulting framework, with its known reference values and calibrated uncertainty, is a clean and useful contribution that is distinct from existing conditional accuracy diagnostics like TARP, SBC, and coverage-based methods.

---

> ### Author Rebuttal · Authors · 2026-03-31
>
> We thank the reviewer for their valuable comments. We address each point below.
>
> ### Sufficiency:
> By examining the sufficiency proof of the modified Mira score considering only the cases where the reference sample falls inside the ball, the reason why the original Mira score is not a sufficient condition for accuracy becomes apparent: when calculating the CDF of $f(K,J)$ on the unit square, two terms must be included: the case $J< K$ (inside the ball) and the case $J>K$ (outside the ball). Assuming $P_N$ is $B(2,1)$ distributed asymptotically, this leads to $2x=\int_0^x dJ f_J(J)+\int_{1-x}^1 dJ f_J(J)$ which does not imply that $f_J$ is uniform. This means a proposal distribution with, for example, higher probability of having the reference sample with normalized rank $J$ from $c$ that's exactly matched by a lower probability of having the reference sample with rank $1-J$ will still produce a $Beta(2,1)$-distributed $P_N$. Since $c$ can be sampled from any distribution, we expect this  to happen only for very pathological cases or very uninformative/unlucky choices of $c$.
>
> ### Choice of distance metric
>
> We re-ran the lensed image experiment (Section 5.3.2) using the same five distance metrics in Appendix C: Euclidean (L2), Manhattan (L1), Chebyshev, Cosine, and Minkowski (p=3). Across L2, L1, Cosine, and Minkowski, MIRA produces nearly identical scores and the same model ranking. The exception is Chebyshev, which is well-known to become degenerate in high-dimensional spaces, making it an unsuitable metric for 12,288-dimensional image vectors regardless of the test used (as noted in Appendix C, MIRA can use any distance metric but becomes less discriminative with an uninformative metric). Excluding this known pathological case, MIRA's conclusions are maintained across common pixel-space metrics.
>
> |Metric|$p_s(x),\sigma_n=2$|$p_s(x),\sigma_n=0.5$|$p_e(x),\sigma_n = 2$|$p_e(x), \sigma_n=0.5$|
> |-|-|-|-|-|
> |L2|0.6489±0.0576|0.5722±0.0651|0.5281±0.0793|0.5002±0.0721|
> |L1|0.6455±0.0601|0.5800±0.0593|0.5263±0.0667|0.5020±0.0670|
> |Chebyshev|0.6670±0.0592|0.5122±0.0592| 0.5378±0.0764|0.5959±0.0669|
> |Cosine|0.6490±0.0564|0.5418±0.0665| 0.5105±0.0833|0.5293±0.0709|
> |Minkowski (p=3)|0.6476±0.0625|0.5734±0.0754| 0.5202±0.0712|0.5068±0.0761|
>
> We also evaluated MIRA in the embedding space of a pretrained frozen ResNet-18 [1]. Despite this encoder never having seen gravitational lensing data as the experiment in 5.3.2 (Fig5), MIRA applied in the embedding space is able to evaluate the quality of the posteriors. It correctly detects that the $p_s(x),\sigma_n=2$ model is correct within sample variance and correctly ranks the rest of the models.
>
> |Posterior| Pixel L2|ResNet-18 Emb L2|
> |-|-|-|
> |$p_s(x),\sigma_n=2$|0.6481±0.0581|0.6836±0.0511|
> |$p_s(x),\sigma_n=0.5$|0.5735±0.0716|0.5712±0.0712|
> |$p_e(x),\sigma_n=2$|0.5174±0.0735|0.5312±0.0756|
> |$p_e(x),\sigma_n=0.5$|0.5115±0.0773|0.5225±0.0712|
>
> MIRA can be applied in any metric space, and its diagnostic power scales with how well the chosen representation captures the relevant distributional differences for the domain. This means domain-specific embeddings can be substituted directly to rank the proposed conditonal distributions.
>
> [1] He, Kaiming, et al. "Deep residual learning for image recognition." Proceedings of the IEEE conference on computer vision and pattern recognition. 2016.

---

> > ### Author Rebuttal · Reviewer_VNeR · 2026-04-03
> >
> > I thank the author for the rebuttal. My questions has been fully resolved and I think a score of 5 is reasonable.

---

### Official Review · Reviewer_9z8D · 2026-03-13

**Soundness:** 4
**Presentation:** 4
**Significance:** 3
**Originality:** 3
**Overall Recommendation:** 5
**Confidence:** 4

**Summary:**

This paper considers a true probabilistic model $M^\star$ and a given candidate $M$.
It aims to assess fidelity of $p(y \mid x, M)$ against the true, unknown $p(y \mid x, M^\star)$ for all $x_i^\star$ in the labeled data.
The paper states that the main distinction of their setting from previous ones is that it assumes that for each $x_i^\star$, we can obtain multiple candidate draws from $p(y \mid x_i^\star,M)$ but only a single ground-truth realization $y_i^\star$ from the true conditional.
To this end, it proposes Mira, a method that measures whether the observed $y_i^\star$ falls into them as often as samples from the candidate model do.

- The key step is that for each fixed $x^\star$, Mira draws a center $c$, then draws a reference point $y_r \sim p(y \mid x^\ast,M)$, and defines a ball around $c$ whose radius is the random distance from $c$ to $y_r$.
- Proposition 3.1 shows that the mass of the candidate-model inside that ball $R$ is uniform.
- Mira sets the null hypothesis to be that $M$ and $M^\star$ assign the same mass $\lambda$ to the region $R$. Through the posterior on $\lambda$, Proposition 3.2 computes the Binomial posterior predictive distribution of $k$, the indicator of whether one more draw after $n$ falls in $R$, yielding Laplace’s rule of succession.
- The statistic of Mira is defined to be $p_N(k \mid n)$ where $N$ is the number of total draws.
- Proposition 3.3 shows that the null distribution of the Mira statistic when $N$ is large is asymptotically $Beta$.
-  Mira score is defined as the mean of the Mira statistic over the $(x^\star, y^\star)$, the region $R$ and draws from the candidate model.
- Corollary 3.4 says that the mean of the statistic, which is the Mira score,  is 2/3, and the variance is 1/18.
Proposition 3.5 says that for independent distributions the Mira score is 1/2.  Together they mean that 2/3 is the reference value for a correct candidate model, and 1/2 is the reference value for a completely uninformative candidate.

**Compliance With Llm Reviewing Policy:**

Affirmed.

**Final Justification:**

My main concern was an error in the proof of one of the propositions. The authors have provided a new proof and I raised my score.

**Key Questions For Authors:**

Please see my question regarding Proposition 3.5 above.

**Limitations:**

yes

**Strengths And Weaknesses:**

This is a well-written paper. The presentation is clear, and the method is developed through a natural, stepwise progression.

The core idea is simple and compelling: it builds on the key observation underlying PQMass (namely, that the counts of samples falling in any collection of disjoint regions follow a multinomial distribution), and extends it to enable testing the fit of a candidate model in the realistic setting in which generative models are evaluated.

My only concern, is that it seems to me that Proposition 3.5 requires independence of random masses $\lambda_n$ and $\lambda_k$, not $y_M \perp y_{M^\star}$.

The proof uses the independence of $\lambda_n \perp \lambda_k$, which are masses of the same random region $R$ under two different conditional distributions. However, $y_M \perp y_{M^\star}$ does not imply independence of $\lambda_n$ and $\lambda_k$. For example, consider the case where the candidate is correct: $p(y \mid x,M)=p(y \mid x,M^\star)$. For a given $x^\star$, we can draw two independent samples $y_M$ and $y_{M^\star}$, but this would yield asymptotic mean $2/3$, not $1/2$.

Additionally, it is stated by the authors in the discussion, Mira does not provide a sufficient condition for correctness. This means that even a score near the null reference value is not enough to conclude that the candidate conditional distribution is correct. Due to the importance of this limitation, I strongly recommend emphasizing this immediately after presenting Corollary 3.4.


------------------------

The authors addressed my concerns and I am therefore raising my score

---

> ### Author Rebuttal · Authors · 2026-03-31
>
> We thank the reviewer for this observation. It is correct: independence of samples $y_{\mathcal{M}}$ ⫫ $y_{\mathcal{M}^\*}$ does not imply independence of the probability masses $\lambda_n$ and $\lambda_k$, and the lambdas cannot actually be indepent as it depends on the same random variable. This correction does not change the intended meaning of the result, which was to characterize the Mira score when the candidate and true distributions are geometrically unrelated. In fact, it leads to a stronger result: we now prove that $1/2$ is a universal lower bound on the Mira score, with equality approached when the candidate and true distributions have disjoint supports. The corrected proposition is:
>
> **Proposition 3.5 (Lower bound on the Mira score).** For any candidate model $\mathcal{M}$,
>
> $$\mu_{\mathrm{Mira}}(\mathcal{M}) \geq \frac{1}{2},$$
>
> with equality approached when the candidate and true distributions have disjoint supports.
>
> Starting from Eq. (38) and substituting $\mathbb{E}[\lambda_n] = 1/2$ (Proposition 3.1):
>
> $$\mu_{\mathrm{Mira}}(\mathcal{M}) = \frac{1}{2} + \frac{N}{N+2}\Big(2\,\mathbb{E}[\lambda_n \lambda_k] - \mathbb{E}[\lambda_k]\Big).$$
>
> Since $N/(N+2) > 0$, we need to show $2\,\mathbb{E}[\lambda_n \lambda_k] - \mathbb{E}[\lambda_k] \geq 0$. For fixed $(x^{\*}, c)$, let $F_D(r) = \int_{d(y,c) \leq r} p(y \mid x^\*, \mathcal{M})\,dy$ and $F_{D^\*}(r) = \int_{d(y,c) \leq r} p(y \mid x^\*, \mathcal{M}^\*)\,dy$ be the radial CDFs. Since $\mathcal{R}$ has radius $r^\* = d(y_r, c)$: $\lambda_n = F_D(r^\*)$ and $\lambda_k = F_{D^\*}(r^\*)$. Using the generalized inverse $F_D^{-1}(t) = \inf\{r \geq 0 : F_D(r) \geq t\}$, we define $T_{x^\*,c}(\lambda_n) = F_{D^\*}(F_D^{-1}(\lambda_n))$, which is non-decreasing (composition of non-decreasing functions). Since $F_D^{-1}(\lambda_n) = r^\*$ a.s. and $\lambda_n \mid x^\*,c \sim \mathcal{U}[0,1]$ for all $(x^\*,c)$ pairs (Proposition 3.1), Chebyshev's integral inequality applied to the non-decreasing functions $\lambda_n$ and $T_{x^\*,c}(\lambda_n)$ gives:
>
> $$\int_0^1 \lambda_n \cdot T_{x^\*,c}(\lambda_n)\,d\lambda_n \geq \frac{1}{2}\int_0^1 T_{x^\*,c}(\lambda_n)\,d\lambda_n,$$
>
> $2\,\mathbb{E}[\lambda_n \lambda_k \mid x^\*, c] - \mathbb{E}[\lambda_k \mid x^\*, c] \geq 0$. Marginalizing over $(x^\*, c)$ yields $\mu_{\mathrm{Mira}}(\mathcal{M}) \geq 1/2$. Equality holds iff $T_{x^\*,c}$ is constant a.e. (Jakubowski, 2021, Theorem 1.2), which with $T_{x^\*,c}(0) = 0$ forces $T_{x^\*,c} \equiv 0$, corresponding to $y^\*$ never being in the region $\mathcal{R}$. However, since $\mathcal{R}$ is a ball in the full space whose radius can be large, $y^\*$ may fall in $\mathcal{R}$ even when the supports are disjoint. Therefore, $1/2$ is approached as the supports become increasingly separated but is not necessarily attained.
>
> ### Sufficiency:
> We thank the reviewer for this suggestion, which we will implement if the paper is accepted. However, there is a small modification to the Mira score that can render it a sufficient condition for calibration, by only considering the cases where the reference sample falls inside the ball. This, however, makes the method more sample inefficient, and we expect it won't be as useful to the general practitioner, hence why it was not included in the original manuscript. We will add it here and can include it as an additional appendix if requested.

---

> > ### Author Rebuttal · Reviewer_9z8D · 2026-03-31
> >
> > Dear authors,
> >
> > I am happy with your new derivation. In fact, I believe your new result makes experimental values much easier to read.
> >
> > As for sufficiency, on my side it will be enough is you emphasizing this after presenting Corollary 3.4. However, personally, I do recommend that you include it in the appendix. Even if not useful for most practitioners, it might be for some.
> >
> > Following you updated result I plan to raise my score.

---

> > > ### Author Response · Authors · 2026-04-03
> > >
> > > We thank the reviewer for their constructive and insightful feedback, and for the increased score.
> > >
> > > We will mention the non-sufficiency directly after Corollary 3.4 to make this point clear earlier in the text.
> > >
> > > Below, we provide the sufficiency proof for the alternative, simulation-intensive score (restricted to cases where the truth falls inside the region). We will include this proof in the appendix.
> > >
> > > ---
> > > ## Sufficiency proof for the simulation-intensive alternative score
> > >
> > > ### Context
> > >
> > > We have shown that $P_N := p(k\\mid n) \\sim Beta(2,1)$ (App A.3) and $\\mu_\\mathrm{Mira}(\\mathcal{M}) \\to 2/3$ in the large $N$ limit (App A.4) are necessary conditions for the proposal and the true distributions to be equal. While neither is a sufficient condition, we can define an alternative Mira score by restricting the Mira score to cases where the true sample falls inside the ball $(k=1)$, and show that the resulting random variable $q_N := p(k=1 \\mid n)$ being $\\text{Beta}(2,1)$ distributed is a sufficient condition for calibration.
> > >
> > > We present this sufficiency proof here as opposed to the main text because restricting to cases where $y^\\star$ falls inside the ball substantially reduces sample efficiency. Since we expect Mira to be most commonly used in sample-limited regimes, the definition in the main text is more practical. However, for settings where data is abundant, and guarantees of accuracy are of the utmost importance, the alternative definition and its sufficiency proof may be preferred.
> > >
> > >
> > > ### Proof derivation
> > >
> > > To start, we reframe the proof in the language of sample ranks. For any given conditional we want to test, for every value of $q_N$ we calculate, we have a set of $N+2$ samples: one samples from the the reference true distribution (the ground truth $y^\\star$ ) and $N+1$ i.i.d. samples from the proposal conditional distribution.
> > > We rank the $N+1$ samples from the proposal distribution based on their distance to the center $c$.
> > > The number of samples inside the ball, $n$, is one minus the rank of $y_r$ among the remaining $N$ samples, since it is the number of samples that are closer to $c$ than $y_r$ (excluding $y_r$ itself). Since the $N+1$ samples are i.i.d. samples from the same distribution, $y_r$ is equally likely to have any rank, meaning that $n$ is a random integer uniformly distributed on $\\{0, ...,N\\}$.
> > > The sample $y^\\star$ is an $(N+2)$-th sample whose position in the rank we want to compare to the region defined by $y_r$.
> > >
> > > We define $J,~K$ as the normalized ranks (by $N+2$) of $y^\\star$ and $y_r$, respectively, which have support on $[0,1]$ when $N \\rightarrow \\infty$. The values of $q_N$ are given by:
> > > \begin{align}
> > >     q_N := p(k=1|n) = K  \\qquad &\\mathrm{since}  J<K .
> > > \end{align}
> > >
> > > First, we prove the necessary direction: if the underlying distributions are the same, then $q_N \\sim Beta(2,1)$.
> > > To find the PDF of $q_N$ we calculate the CDF $F(x)=P(q_N\\leq x)$ using a  geometric argument,  integrating the area in $(J,K)$ space where $y^\\star$ is inside the ball. This is the area of the triangle defined by $0<K<J$ and $K<x$, which has area $x^2/2$. Normalizing to make this a CDF on [0,1], this means the CDF of $P(q_N)$ is $F(x) = x^2$, and
> > > \begin{align}
> > >     P(q_N)= \\frac{d(q_N^2)}{dq_N} = 2q_N \\qquad \mathrm{for} \\quad q_N\, \\in \, [0,1]\, ,
> > > \end{align}
> > > which is the PDF of $Beta(2,1)$.
> > >
> > > Now, to prove the sufficiency condition, we assume $q_N \\sim Beta(2,1)$. We define $K$ and $J$ as above, and let $f(K,J)$ be their joint distribution on $[0,1]$. To show that the reference and the proposal distribution are identical, it is enough to show that $f(K,J)$ is uniform on the unit square, that is, $K$ and $J$ must be uniformly distributed on $[0,1]$. As before, we know that
> > > \begin{align}
> > >     q_N= p(k=1|n) = K \\qquad & \\mathrm{ since } J<K
> > > \end{align}
> > >
> > > Since $q_N \\sim Beta(2,1)$  by assumption, the CDF must be $F(x)=x^2$.
> > > As before, this can also be calculated from $f(K,J)$ by integrating the probability that $J<K$ and $q=K\leq x$ for a given value of $x$:
> > > \begin{align}
> > >     F(x) = \\int_0^x dK \\int_0^K dJ \\left[ f(K, J) \\right] = x^2.
> > > \end{align}
> > > Differentiating with respect to $x$ we obtain, using the fundamental theorem of calculus:
> > > \begin{align}
> > >     \frac{d}{dx}F(x)=2x=\\int_0^x dJ \\left[f(x,J)\\right].
> > > \end{align}
> > > We know the normalized rank $K$ must be uniform, since it is the rank of $y_r$ among $i.i.d.$ samples from the same distribution. Therefore, $f(K,J) = f_K(K)f_J(J)=C\,f_J(J)$, with $C$ a constant, and we find:
> > > \begin{align}
> > >     2x &=C \\int_0^x dJ \\left[f_J(J)\\right]= C\\left(F_J(x)-F_J(0)\\right)= CF_J(x),
> > > \end{align}
> > > where $F_J$ denotes the CDF of $f_J(J)$. Since this must hold for all $x$, we have that $f_J(x)$ must be uniform on $[0,1]$ for any value of $K$, and therefore the joint $f(J,K)$ must be uniform on the unit square, which completes the proof.

---

### Decision · Program_Chairs · 2026-04-30

**Decision:**

Accept (spotlight)

**Comment:**

This paper presents MIRA, a geometric, random-region generalisation of PIT/SBC that allows for assessment of conditional distribution accuracy in generative models.

Strengths: the reviewers agree that the method is elegant with clear derivations, the theory is correct (after the correction given in the rebuttal), the MIRA score itself is relative interpretable, and the experiments show it can rank models when TARP/SBC/LC2ST can struggle, including scaling to high dimensions.  The applicability of the work to a real problem is also clear; I can really see people using this approach in practice with useful benefits.

Weaknesses: after the discussions in the rebuttal I believe there are two clear weaknesses outstanding: 1) the conditions provided are necessary but not sufficient conditions for the distributions to be the same so there are not guarantees of correctness even when hte MIRA comes out at the expected value.  2) the relationship with PIT/SBC is underdeveloped and not fully discussed; the novelty relative to this work is not really as strong as the paper would suggest at present and prior art is not given sufficient credit.

Overall, this is a very solid paper and I am in agreement with the unanimous decision of the reviewers that it should be accepted.  In particular, I do not think either of the weaknesses above are fatal, with the former already explicitly acknowledged and a natural limitation of the otherwise useful approach, and the latter something that can be partly addressed in the revision (in terms of positioning relative to the previous work) and the work still meeting the required bar for novelty in spite of this.

Because of the potential usefulness of the work and relatively broad appeal, I think the work is potentially suitable for an oral, though I would not push overly hard for this because of the close relationship with existing PIT work.